# Dynamic modulation of the lipid translocation groove generates a conductive ion channel in $Ca^{2+}$-bound nhTMEM16

George Khelashvili [1,2,6]*, Maria E. Falzone[3,6], Xiaolu Cheng [1], Byoung-Cheol Lee[4,5], Alessio Accardi [1,3,5]* & Harel Weinstein [1,2]*

Both lipid and ion translocation by $Ca^{2+}$-regulated TMEM16 transmembrane proteins utilizes a membrane-exposed hydrophilic groove. Several conformations of the groove are observed in TMEM16 protein structures, but how these conformations form, and what functions they support, remains unknown. From analyses of atomistic molecular dynamics simulations of $Ca^{2+}$-bound nhTMEM16 we find that the mechanism of a conformational transition of the groove from membrane-exposed to occluded from the membrane involves the repositioning of transmembrane helix 4 (TM4) following its disengagement from a TM3/TM4 interaction interface. Residue L302 is a key element in the hydrophobic TM3/TM4 interaction patch that braces the open-groove conformation, which should be changed by an L302A mutation. The structure of the L302A mutant determined by cryogenic electron microscopy (cryo-EM) reveals a partially closed groove that could translocate ions, but not lipids. This is corroborated with functional assays showing severely impaired lipid scrambling, but robust channel activity by L302A.

[1] Department of Physiology and Biophysics, Weill Cornell Medical College of Cornell University, New York, NY 10065, USA. [2] Institute for Computational Biomedicine, Weill Cornell Medical College of Cornell University, New York, NY 10065, USA. [3] Department of Biochemistry, Weill Cornell Medical College of Cornell University, New York, NY 10065, USA. [4] Research Group for the Neurovascular Unit, Korea Brain Research Institute, Daegu, Republic of Korea. [5] Department of Anesthesiology, Weill Cornell Medical College of Cornell University, New York, NY 10065, USA. [6]These authors contributed equally: George Khelashvili, Maria E. Falzone. *email: gek2009@med.cornell.edu; ala2022@med.cornell.edu; haw2002@med.cornell.edu

on transport and phospholipid (PL) scrambling are key physiological functions of TMEM16 membrane proteins[1]. Some family members, such as TMEM16A and TMEM16B, are $Ca^{2+}$-activated $Cl^-$ channels important for ion transport in olfaction, phototransduction, smooth muscle contraction, nociception, cell proliferation and control of neuronal excitability[2]. Others, such as TMEM16E, 16F, and 16K, are PL scramblases and non-selective ion channels[1] are implicated in genetic disorders of blood, muscle, bone, and brain[3–6]. The diversity of organs and tissues impacted by the impaired activity of these scramblases underscores their key roles in a wide range of physiological processes.

The breakthrough X-ray structure of a fungal TMEM16 homologue, nhTMEM16[7], revealed that TMEM16 proteins are homo-dimers. Each subunit is composed of 10 transmembrane (TM) segments, with structured termini on the cytoplasmic side and featuring a polar cavity (groove) that faces the lipid membrane and serves as the permeation pathway. These grooves span the entire thickness of the membrane and are lined by TMs 3–7. TM6 and TM7 also participate in the formation of the two regulatory $Ca^{2+}$ binding sites located within the membrane[7]. The structure of the TMEM16A channel[8–10] revealed that this architecture is conserved between both TMEM16 subtypes with differences only in the groove region. This suggested that subtle rearrangements of the groove are sufficient to enable transport of diverse polar substrates. Structural, functional and computational experiments[11–15] led to the proposal of a lipid scrambling mechanism by the TMEM16 scramblases consistent with the card-reader model[16]. However, the mechanistic basis for ion permeation through a TMEM16 protein remains poorly characterized, in large part due to lack of understanding of how the diverse structural states assumed by TMEM16s relate to their functional properties.

Recently, we and others[8,10,17–19] showed that the binding of $Ca^{2+}$ to TMEM16 proteins alters the groove conformation. Under apo conditions, the $Ca^{2+}$ binding sites of the TMEM16 scramblases are disrupted by repositioning of the intracellular half of TM6[9,10,17–19], the extracellular portion of TM4 bends and interacts with TM6, occluding the groove from the membrane (except in nhTMEM16 in detergent[18]). In this conformation, termed here $Ca^{2+}$-free closed, the internal groove diameter of <1 Å[19] renders this state transport-incompetent for both lipids and ions. $Ca^{2+}$ binding induces a straightening of TM6 to form the $Ca^{2+}$ sites, and TM4 takes on distinct conformations in which the groove can be exposed to the membrane or occluded from the membrane. Specifically, TM4 can straighten out to reveal the hydrophilic interior of the groove to the lipid membrane (membrane-exposed conformation)[7,18–20]. This conformation was initially proposed to be conductive for lipids, although the internal diameter of the groove is too narrow to allow permeation of lipid headgroups. We showed recently that in nhTMEM16 the constrictions narrowing the groove can be released to generate a lipid-conductive conformation upon concerted dissolution of a polar interaction network tethering together TM3, TM4 and TM6[21]. Interestingly, in the $Ca^{2+}$-bound nhTMEM16, TMEM16K and TMEM16F, TM4 can bend to a conformation similar to that in $Ca^{2+}$-free condition, thereby occluding the groove from the membrane ($Ca^{2+}$-bound closed conformation) and constricting its internal diameter[17,18,20].

Remarkably, in the $Ca^{2+}$-bound nhTMEM16 and TMEM16F structures determined with cryo-EM in lipid nanodiscs[17,18] a third conformation was identified, where a bent TM4 contacts TM6, occluding the groove from the membrane and inhibiting scrambling, but is repositioned away from TM5, widening the internal diameter of the groove. This intermediate conformation resembles the conformation of the ion pore of the $Ca^{2+}$-bound TMEM16A, which is non-permissive to both lipids and ions due to its small internal radius[8–10]. This conformation is likely a

stable state as it encompasses ~30% of the cryo-EM particles of nhTMEM16[18] and nearly all particles of mTMEM16F[17,22]. Notably, the groove in the intermediate conformation is wider than the $Ca^{2+}$-bound TMEM16A pore, leading to the proposal that the intermediate groove conformation might mediate ion transport[17,18]. Importantly, the molecular mechanisms underlying the formation and stability of this intermediate state remain unexplored, and it is unknown if this intermediate structure represents an ion-conductive state. Here we combined atomistic molecular dynamics (MD) simulations with structure/function experiments to investigate the mechanisms regulating the conformational transitions of the groove in $Ca^{2+}$-bound nhTMEM16 that generate, and stabilize, an ion-conducting state.

In our computational studies of nhTMEM16[21] we had observed rare events of spontaneous transitions of the groove from $Ca^{2+}$-bound membrane-exposed to the intermediate conformation. In these cases, both the headgroup and the tails of lipids penetrate the groove as TM4 repositions closer to TM6 to yield a conformation resembling the nhTMEM16 intermediate state. Here, we expand the set of MD trajectories to sample these rare events more frequently and identify the sequence of conformational transitions that convert the $Ca^{2+}$-bound nhTMEM16 groove from the lipid-conductive to the intermediate state. We learn how disruption of the coupled dynamics of TM4 and TM3 facilitates structural rearrangements involving TM4 that lead to conformations where the groove is closed to lipids, but remains sufficiently wide for ion permeation. Mutations designed to disrupt the hydrophobic interactions between TM3 and TM4, severely impair scrambling but maintain significant channel activity. Our cryo-EM studies of one of the mutants, L302A, demonstrate that this mutation stabilizes the groove in a conformation that verifies the computational prediction, and is similar to the intermediate state seen in nhTMEM16[18]. The L302A groove is occluded to the membrane, and therefore scrambling-incompetent, but forms an open, continuous TM pore that can allow ion movement.

## Results

**Intermediate state of the groove in $Ca^{2+}$-bound nhTMEM16.** We recently reported on rare scrambling-incompetent conformational states observed in our MD simulations of the $Ca^{2+}$-bound nhTMEM16[21] in which the groove is structurally similar to the $Ca^{2+}$-bound intermediate conformations of TMEM16 scramblases and channels[9,10,17,19]. These conformations, favored by the insertion of a lipid tail into the groove, are characterized by a dehydrated translocation pathway and by a shortening of the distance between the central portions of TM4 and TM6. To increase the sampling of such rare states, we produced an additional set of long atomistic MD simulations of the same system (see Supplementary Table 1 and Methods) for an enhanced trajectory complement of ~34 μs where we observe these events multiple times, allowing us to elucidate the underlying mechanisms. From the tICA representation of the enhanced trajectory complement (see "Methods"), and the discretization of the resulting 2D tICA space into microstates we find the same microstate regions and transitions identified previously to describe lipid permeation[21] (see Supplementary Note 1).

The increased sampling of these intermediate conformations is evidenced in Fig. 1a by the higher density of microstate 3. In 5/12 nhTMEM16 subunits in the WT1-WT6 simulations, and 2/48 subunits in the WT$^{ensemble}$ simulations (see Supplementary Table 1, and text below) this conformation was long lived. In the simulation frames collected in microstate 3, the $C_\alpha$-$C_\alpha$ distance between V337 (on TM4) and V447 (on TM6) is ~7–8 Å, which corresponds to a major reduction (by ~4–5 Å) of that distance

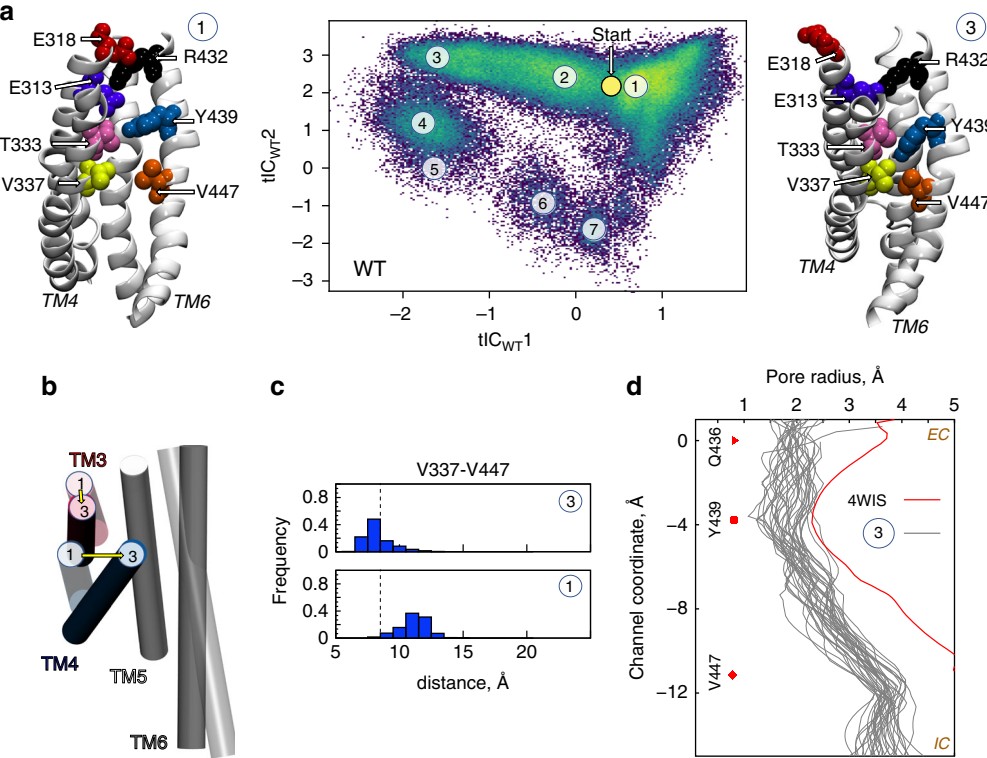

**Fig. 1** Structural characteristics of states in the lipid translocation mechanism. **a** Central panel: 2-D landscape representing all the trajectories of the WT nhTMEM16 (Supplementary Table 1) mapped with the tICA transformation in the space of the first two tICA eigenvectors (tIC$_{WT}$ 1 and tIC$_{WT}$ 2; see "Methods"). The lighter shades (green to yellow) indicate the most populated regions of the 2D space. The dynamics of the two subunits of the protein in each trajectory were analyzed separately. Microstates representing the most populated states in these simulations are indicated by the numbered circles and represent various stages in the lipid translocation process. Representative structures of microstates 1 and 3 are shown in the surrounding snapshots. In these models, TM4 and TM6 are labeled; the relevant groove residues appear in space-fill representations and are labeled (see also Supplementary Fig. 1A). The location of the initial conformation of the system (the 4WIS model) is indicated by the yellow circle marked Start. **b** Structural comparison of the position of groove helices TM3-TM6 (in cartoon) in microstates 1 and 3. Note the major repositioning of TMs 3 and 4 from microstate 1 (light gray) to microstate 3 (dark gray). **c** The probability distributions of the C$_\alpha$-C$_\alpha$ distance between V337 and V447 residues in the ensemble of conformations from microstates 1 and 3. The vertical dashed lines represent the 8.5 Å distance cut-off used to define the occluded conformation of the groove. **d** The pore radius as a function of position along an axis perpendicular to the membrane (channel coordinate) for the X-ray structure of nhTMEM16 (PDBID 4WIS, red line), and for selected structures from microstate 3 (gray lines) representing 34 frames (separated from each other by at least 10 ns time interval) from the last 500 ns of a 15 μs MD simulation (WT6 in Supplementary Table 1). The EC and IC ends of the pore are indicated; Z = 0 Å corresponds to the location of the C$_\alpha$ atom of residue Q436 (red triangle symbol), the dot marks the position of the C$_\alpha$ atoms of residue Y439, and the diamond symbol that of V447. The calculations were performed with HOLE (http://www.holeprogram.org/).

compared to the membrane-exposed or lipid-conductive conformations (Supplementary Fig. 3B: TM4-TM6 MID Histograms). At this C$_\alpha$-C$_\alpha$ separation, the two residues are in direct contact (minimal distance of <2.5 Å), as the middle region of TM4 is repositioned towards TM6 occluding the groove from the membrane (see structural representation of microstate 3 in Fig. 1a). Additionally, the extracellular half of TM3 moves away from TM6 and TM5 (Fig. 1b and Supplementary Fig. 3B), leading to an overall contraction of the groove compared to the membrane-exposed conformation (Fig. 1b). These rearrangements isolate the hydrophilic interior of the groove from the membrane and render it non-conductive for lipids. In the ensemble of conformations comprising microstate 3, the groove is structurally more similar to that seen in the Ca$^{2+}$-bound intermediate of nhTMEM16 (6QMA) and mTMEM16F (6QPC), than to the Ca$^{2+}$-bound or Ca$^{2+}$-free closed conformations[17,18] (see Supplementary Table 3).

**Dynamic rearrangements of TM3 and 4 close the lipid pathway.** Transition to the intermediate states populating microstate 3

is promoted by sustained interactions of a groove-penetrating lipid tail with V337 in TM4 and V447 in TM6 (Fig. 2a–c; Supplementary Fig. 4A–C, G–I, P–R; Supplementary Movie 1). Many instances of lipid tail penetration into the groove with concomitant reduced hydration of the groove were also observed in the conformations corresponding to microstates 4 and 7 (Supplementary Fig. 5) which produce the shoulder (at low counts) in the overall water distribution plot (Fig. 3b), but lipid tail penetration was a minor occurrence in the other microstates.

In the longest (15 μs) MD simulation, WT6, the lipid insertion and concomitant occlusion of the groove from the membrane occurs in both protomers (Fig. 2a–c, Supplementary Fig. 4B, E, H). In one case the lipid tail coordinates V337/V447 for most of the trajectory (Supplementary Fig. 4H) and the groove closure occurs during the last stages of the simulation while the tail is still inserted into the groove (Supplementary Fig. 4B). In the other protomer, the groove remained in the intermediate state for the final third of the simulation (Fig. 2b) although the penetrating lipid tail had already left the groove (Fig. 2c, d). This suggests that while the rearrangements of TM4 and TM6 are promoted by hydrophobic contacts between the inserted lipid tail and V337/V447, once the

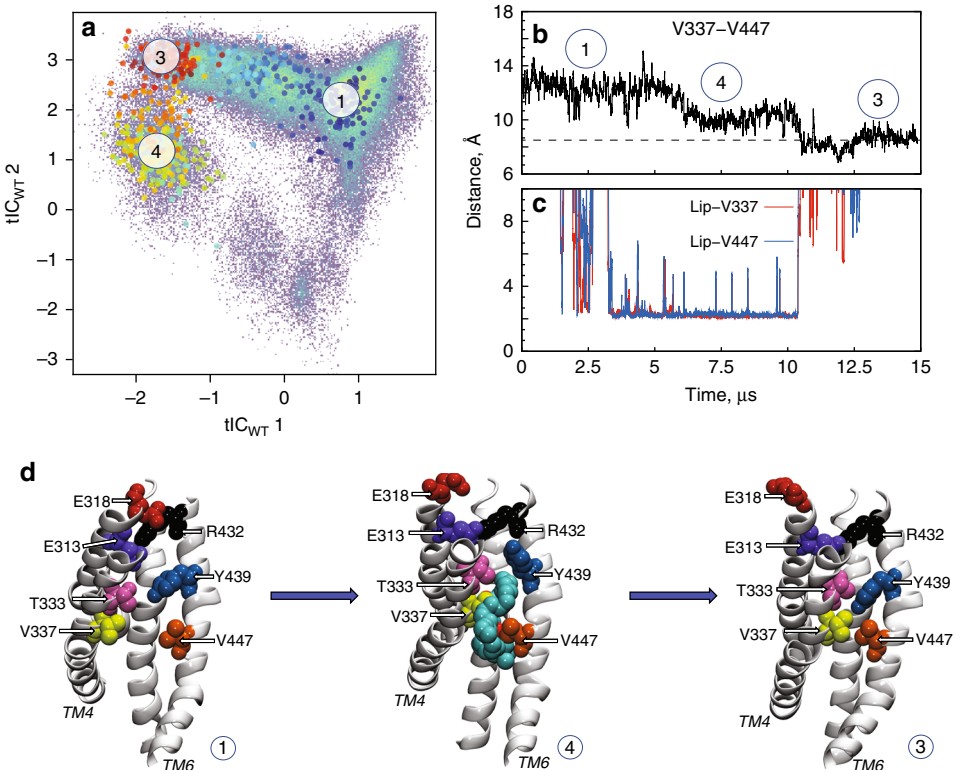

**Fig. 2** The time evolution of an MD trajectory shows that occlusion of the groove is triggered by lipid tail insertion. **a** Projection (large colored dots) of the 15μs-long WT6 MD trajectory calculated for one of the nhTMEM6 subunits on the 2D tICA landscape from Fig. 1. The colors of the large dots indicate the time-frames in the evolution of the trajectory: darker colors (blue, cyan) are indicating the initial stages of the simulation, lighter colors dots (yellow, green) correspond to the middle part of the trajectory, and red shades show the last third of the trajectory. In this representative trajectory the system is seen to have evolved from microstates 1 to 4, and to 3. **b** The evolution of the $C_\alpha$-$C_\alpha$ distance between residues V337 and V447 as a function of time in the WT6 trajectory. **c** Time evolution of the minimum distance between the lipid tail penetrating the groove and residues V337 (in red) and V447 (in blue). **d** Structures of representative snapshots showing the gradual occlusion of the groove in going from microstate 1 to 4 and to 3. The color code in (**d**) is the same as in Fig. 1a; TM4 and TM6 are labeled. Note that the lipid tail present inside the groove in microstate 4 is absent in the ensemble of states in microstate 3, suggesting that once the occluding configuration was established, the lipid tail is no longer necessary for its relative stability

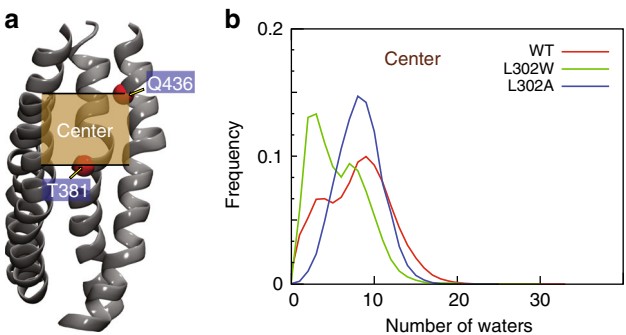

**Fig. 3** Hydration of the central region of the groove in the WT and mutant nhTMEM16. **a** A snapshot of the nhTMEM16 (TMs 3–6) from the WT2 simulation showing the location of the central region of the groove (the tan shaded rectangle). The thick black lines surrounding the shaded rectangle indicate the z-axis positions of the $C_\alpha$ atoms of residues T381 and Q436 (red spheres). **b** The frequency of finding a specific number of water molecules in the central region of the groove calculated from the full set of the WT nhTMEM16 (red), L302A (blue), and L302W (blue) trajectories (see "Methods" for the definition of water distribution in the groove and its different regions). For each construct, the data shown are averaged over the two protomers of the protein

resulting occluded conformation of the groove is established it remains stable also in the absence of the lipid tail. In one of the nhMEM16 subunits from the WT1 simulation, the lipid tail remains in the groove for nearly the entire 3 μs trajectory (Supplementary Fig. 4G) with the groove occlusion occurring at ~0.75 μs into the trajectory being sustained until the end (Supplementary Fig. 4D). In all other instances, groove closure occurs either at the end of the simulation (Supplementary Fig. 4F), or is transient (Supplementary Fig. 4M–O). The 2D tICA landscapes in Supplementary Fig. 4 show that all these trajectories progress towards microstate 3. Importantly, in all cases, the hydrophobic tails of a single lipid molecule are found to coordinate V337 and V447 prior to the occlusion event (Supplementary Fig. 4G–I, P–R, S–T).

Rearrangements of TM4 that lead to groove occlusion require the reorganization of interactions that can stabilize various conformations available to the flexible portions of this TM. Hydrophobic interactions between TM4 and TM3 that keep the groove open are strengthened by I343 and L347 in TM4, and L302 in TM3 (Fig. 4a). Figures 4b, S6A, and S6C show how L302 engages with the I343/L347 pair in the trajectories of WT and of the A385W mutant of nhTMEM16[21]. In conformations in which the extent of lipid tail penetration was the strongest (microstates 3 and 7, Supplementary Fig. 5A), the interaction of L302 and I343 was weakened and they were found more often at larger, non-interacting separations (minimal L302 to I343 distance > 3.5 Å)

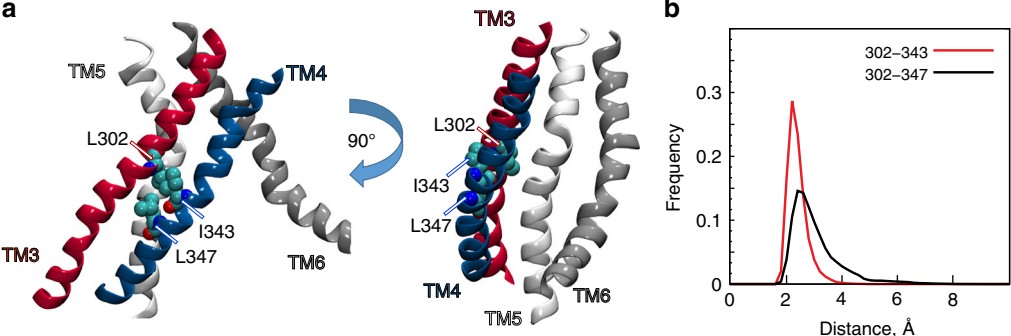

**Fig. 4** TM3 and TM4 in nhTMEM16 are locked in hydrophobic interactions mediated by the L302/I343/L347 triad of residues. **a** Two views, related by 90° rotation around the membrane normal, of the groove region in the WT nhTMEM16 protein, show the locations of L302, I343, and L347 (in space fill and labeled). Helices TM3-TM6 that line the groove are labeled, and colored red, blue, white and gray, respectively. **b** Histograms of minimal distance between residue pairs L302 and I343 (in red), and L302 and L347 (in black), in the simulation trajectories of WT nhTMEM16 (see also Supplementary Fig. 6)

(Supplementary Fig. 6B). We reasoned, therefore, that disrupting the L302/I343-L347 interactions might reduce the coupling between TM3 and TM4, thereby facilitating the rearrangement of TM4 that leads to closure of the groove.

**Identification of mutants enabling enhanced TM4 dynamics.** We previously reported that the L302W mutation resulted in >100-fold decrease in scrambling by nhTMEM16[21]. We thus investigated how L302W affects the conformation of the groove in 7 μs-long MD simulations of nhTMEM16–L302W (Supplementary Table 1; Supplementary Fig. 7, panels A, B and D). We find the large Trp side-chain at position 302 to lead to substantial bending of TM4 that separates the middle regions of TM3 and TM4 (Fig. S8A and D) to yield a larger hydrophobic area of the groove (Supplementary Fig. 8B). This groove becomes progressively filled by lipid tails (Supplementary Fig. 8C, E) causing dehydration of the region (Fig. 3) and a scrambling-incompetent state, in agreement with the experimentally determined phenotype[21]. However, the V337-V447 interaction that allows the middle regions of TM4 and TM6 to come closer and occlude the groove, does not occur in the L302W construct, most likely because it is prevented by the bulging of TM4 (Supplementary Fig. 8A–D). We therefore tested computationally whether a small residue substitution, L302A, would disrupt the L302/I343-L347 interactions holding TM3 and TM4 together while permitting the V337-V447 association and thus groove closure.

Extended MD simulations of the L302A construct reveal that states in which V337/V447 interactions bring together the middle regions of TM4 and TM6 are highly populated. Applying the same tICA analysis (and same parameters) as for the WT, to the 3 μs-long MD trajectories of the L302A system (Supplementary Table 1), we projected the L302A trajectory frames onto the 2D tICA landscape of the WT protein. The superposition of the trajectories for the two systems, and the structural representation of the principal microstates (a–c) (Fig. 5) underscore the high population density of microstate a (Supplementary Fig. 9), in which the TM4-TM6 interaction is mediated by V337/V447. Notably, the states sampled by the mutant are structurally similar to the nonscrambling states of the WT either because of a closed EC gate[21] (microstate b in Fig. 5c and S9, corresponding to a membrane-exposed conformation), or due to rearrangement of TM4 that closes the groove because of the lipid tail insertion (microstates a and c in Fig. 5 and S9, corresponding to an intermediate conformation). Remarkably, the computationally identified conformation of the L302A groove resembles the intermediate conformation of nhTMEM16, as the root-mean-square-deviation (RMSD) of the backbone atoms of groove-lining TMs (TM3–6) is

~1.8 Å (Fig. 5d). The corresponding RMSDs between the X-ray model of nhTMEM16 (PDBID: 4WIS) and either the intermediate structure of nhTMEM16, or the L302A mutant, are larger (3 Å).

In the two extensive simulations of the L302A dimer, three out of the four subunits (Supplementary Fig. 10C, D, F, H), prefer the intermediate conformation (Fig. 5, microstates a and c). As seen in the WT case, this conformation is favored by the lipid tail insertion (Supplementary Fig. 10E–F) but persists after the tails disengage from the groove (Supplementary Fig. 10A–D). The fourth protein subunit of the L302A mutant still experienced lipid tail insertions (Supplementary Fig. 10G) but does not visit the intermediate states (microstate b in Fig. 5 and Supplementary Fig. 9; Supplementary Fig. 10H).

Analysis of the full complement of MD simulation trajectories supports a coherent mechanistic hypothesis of structure-based dynamic modulation of the functional properties of the nhTMEM16 scramblase by lipids—from the lipid-dependent gating of the electrostatic network of interactions at the EC end of the molecule[21], to the modes of function-determining rearrangements of the lipid pathway in the groove as a result of changes in specific TM interfaces. The predictions prompted experimental probing to establish their validity and examine the implications of the corresponding mechanistic inferences.

**Mutations at the TM3/TM4 interface disrupt lipid scrambling.** The three interfacial residues—L302 on TM3, and I343 and L347 on TM4 of nhTMEM16 were mutated to alanine and the purified mutants evaluated for scrambling and ion channel activity in liposomes[21]. The L302A and the I343A mutants have severely impaired lipid scrambling (>100-fold reduction in the scrambling rate constants: Fig. 6a, b; Supplementary Fig. 11A). The L347A mutant has a less severe phenotype, with a ~10-fold reduction in scrambling (Fig. 6a, b; Supplementary Fig. 11A). Simultaneous alanine substitutions of both TM4 residues (I343A/L347A) result in complete loss of activity, with a >1000-fold reduction (Fig. 6a, b; Supplementary Fig. 11A). All mutants reconstitute in liposomes with comparable efficiency, indicating that the decreased scrambling activity does not reflect impaired stability and/or reconstitution efficiency (Supplementary Fig. 11B, C). These results substantiate the role of the hydrophobic lock at the TM3/TM4 interface in maintaining the membrane-exposed and lipid-conductive conformations of the lipid pathway.

**Mutations at the TM3/TM4 interface maintain channel activity.** In contrast to the drastic reduction of scrambling activity, the single alanine substitutions have only minor effects on the Ca²⁺-dependent non-selective channel function of nhTMEM16

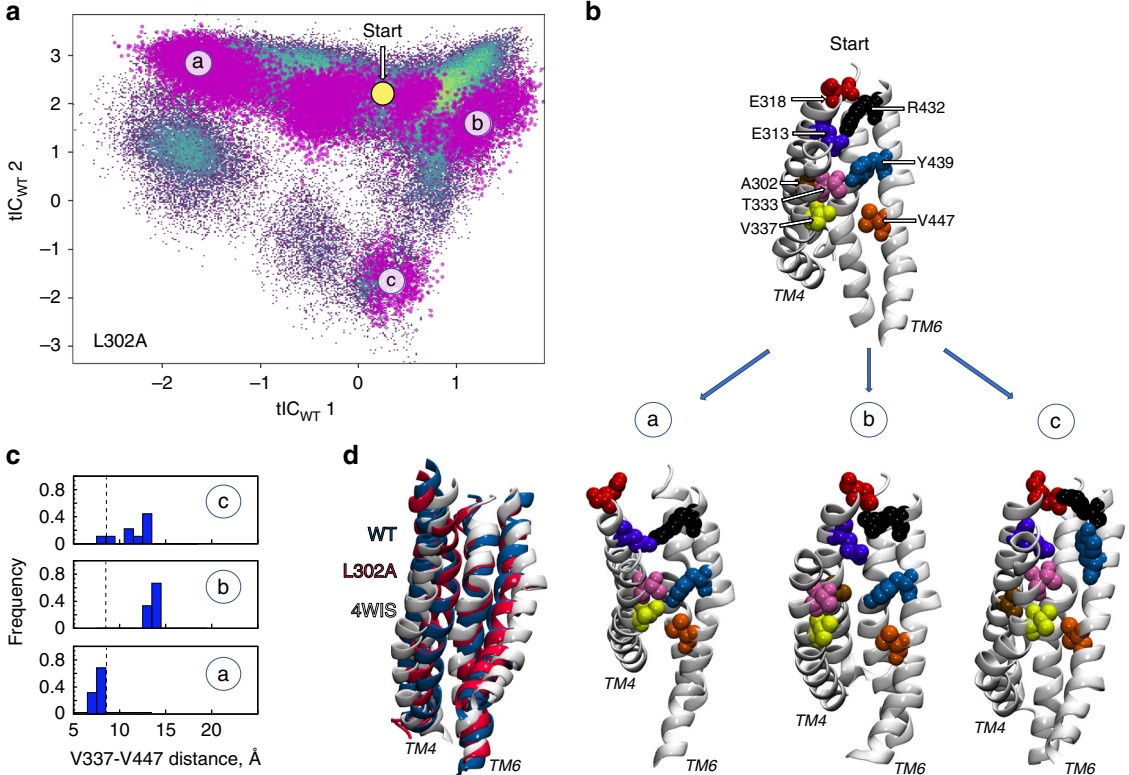

**Fig. 5** Structural characteristics of occluded groove conformation in L302A nhTMEM16. **a** Projection of all the L302A trajectories (purple symbols) onto the 2D tICA landscape of the wild-type system from Fig. 1a. The dynamics of the two subunits of the protein in each trajectory were considered separately in the analysis. The location of the initial conformation of the system (the 4WIS X-ray model) is indicated on the 2D tICA landscape by the yellow circle marked Start. From this configuration, the L302A system evolved towards the microstates denoted by a–c described in the other panels (see also Supplementary Fig. 9). **b** Structural representation of the initial conformation of the L302A system and of the representative conformations of microstates a, b, and c. In these structural models, the relevant groove residues appear in space fill representations, and are labeled. **c** The probability distributions of the $C_\alpha$-$C_\alpha$ distance between V337-V447 residues in microstates a, b, and c. The vertical dashed lines represent the 8.5 Å distance cut-off used to define the occluded conformation of the groove. **d** Structural superpositions with respect to the X-ray structure of the nhTMEM16 (PDBID: 4WIS; in white), of the groove regions (helices TM3-TM6) from the occluded groove conformation in the WT nhTMEM16 simulations (in blue) and from the L302A simulations (in red). The simulated structures of the wild type and the L302A nhTMEM16 are represented by the centroids of the respective ensemble of conformations (i.e., for the wild type—microstate 3 in Fig. 1a; for the L302A—microstate a in Fig. 5a, b)

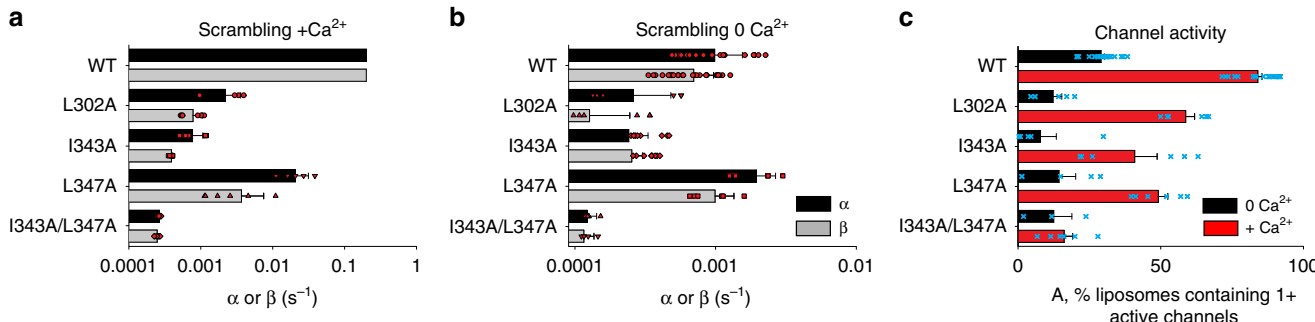

**Fig. 6** Functional consequences of mutations at the TM3/TM4 interface. **a, b** Quantification of forward (α) and reverse (β) scrambling rate constants for WT and mutant nhTMEM16 in the presence (**a**) and absence of $Ca^{2+}$ (**b**) determined by fitting fluorescence traces to Eq. 1 ("Methods"). For the WT protein in the presence of $Ca^{2+}$ the rate constants could not be determined and were constrained to be 0.2 s$^{-1}$ as previously described[21,52]. **c** Quantification of the fraction of liposomes containing at least one active ion channel in the presence (red) and absence (black) of $Ca^{2+}$ using Eq. 3 ("Methods"). Error bars represent S.D. and the values of individual experimental replicates are indicated as symbols. $n = 4$–20 from 2 + independent preparations

measured with the flux assay. Indeed, all single mutants retained robust $Ca^{2+}$-dependent ion transport, with < 2-fold changes in the fraction of liposomes containing at least one active channel (Fig. 6c). Only the double mutation I343A/L347A caused a nearly complete loss of channel activity (Fig. 6c), suggesting that this construct favors a conformation of the nhTMEM16 groove that is non-conductive to both ions and lipids. Thus, the L302A mutant loses most of its lipid scrambling activity while maintaining significant channel function. This suggests that the mutation stabilizes an open-channel conformation of the groove, consistent

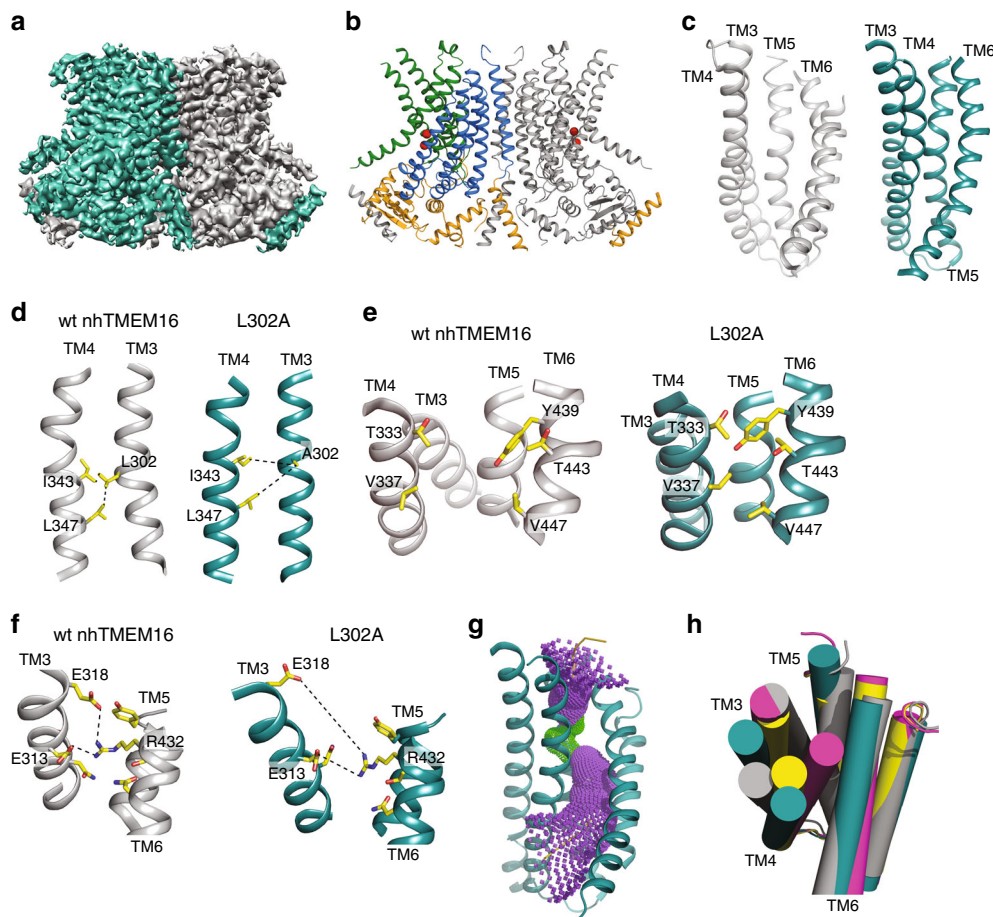

**Fig. 7** Structure of nhTMEM16 L302A in lipid nanodiscs. **a** Masked cryo-EM density maps of L302A nhTMEM16 in the presence of 0.5 mM $Ca^{2+}$. One monomer is shown in gray, the other is teal. **b** Atomic model of L302A nhTMEM16. One monomer is gray, in the other the cytosolic domain is orange, the permeation pathway is green, and the remainder of the protein is blue. $Ca^{2+}$ ions are shown as red spheres. **c**–**f** Structural comparison of the lipid pathway of wt (4WIS, left panels, gray) and L302A (right panels, teal) nhTMEM16. The color scheme is the same throughout the figure. **c** The lipid pathway (TM3-TM6) of the L302A mutant and WT (4WIS) viewed from the plane of the membrane. **d**–**f** Close up views of the structural rearrangements induced by the L302A mutation at the lipid pathway. The protein backbone is shown in ribbon representation and important residues are shown as sticks and colored in CPK yellow. **d** L302A disrupts the TM3/TM4 interface. Dashed lines indicate the distances between L302 on TM3 and I343/L347 on TM4. The distances respectively change from 3.7 to 7.4 Å and from 5 to 7.2 Å indicating that TM4 moves away from TM3. **e** Closure of the pathway to the membrane. The movement of TM4 towards TM6 enables the packing of the side chains of T333/V337 on TM4 and Y349/T443/V447 on TM6 to form a steric barrier to lipid entry. **f** Opening of the extracellular gate. The rearrangements induced by the L302A mutation cause the disengagement of E313 and E318 on TM3 from R432 on TM6. The R432-E313 distance increases from 4.9 Å (WT) to 8.9 Å (L302A) while the R432-E318 distance increases from 4.5 Å (WT) to 12.3 Å (L302A). **g** The diameter of the L302A groove was estimated using the HOLE program[50]. Purple denotes areas of diameter $d > 5.5$ Å and green areas where $2.75 < d < 5.5$ Å. **h** Top view of the lipid pathway. TM3–6 are shown as cartoon helices. Gray: $Ca^{2+}$-bound open nhTMEM16 (4WIS); pink: $Ca^{2+}$-bound closed nhTMEM16 (6QMB); yellow: $Ca^{2+}$-bound intermediate (6QMA); teal: L302A nhTMEM16

with our computational prediction that the groove of the L302A mutant adopts a conformation that is hydrated, occluded from the membrane, and cannot scramble lipids (Fig. 3).

**The L302A groove is closed to lipids but open to ions**. To relate the mechanistic hypothesis to structural determinants, we used single particle cryo-EM to determine the structure of the L302A mutant reconstituted in lipid nanodiscs in the presence of 0.5 mM $Ca^{2+}$, to a resolution of 4.0 Å (Fig. 7a, Supplementary Figs. 12–13). The core TM region is well resolved, with areas of higher resolution, while the cytosolic domains and some extracellular loops are less well-defined (Supplementary Fig. 12F, Supplementary Fig. 14). The L302A mutant adopts the canonical bi-lobed TMEM16 fold, with TM1-2, TM5-10 and the cytosolic N- and C-termini occupying nearly the same positions as in the structures of $Ca^{2+}$-bound

WT nhTMEM16 (Fig. 7b, Supplementary Fig. 12F). Both $Ca^{2+}$ binding sites are occupied, as indicated by strong density for the bound ions visible in the density map as well as in an omit difference map, calculated between experimental data and simulated map not containing $Ca^{2+}$ (Supplementary Fig. 14I, J)[10,19]. The nhTMEM16 L302A/nanodisc complex is bent along the dimer cavity (Supplementary Fig. 12I), consistent with computational predictions[13,23] and recent structural studies[18,19].

Comparison of the L302A and wild-type nhTMEM16 (PDBID: 4WIS) structures shows that the mutation stabilizes a $Ca^{2+}$-bound conformation with the groove closed to the membrane due to a movement of TM4 towards TM6 (Fig. 7c). The predicted disruption of the interaction between TM3 and TM4 is validated by the observed increases in distance between A302 on TM3 and I343/L347 on TM4 from 3.7 to 7.4 Å and from 5.0 to 7.2 Å (Fig. 7d). This causes the repositioning of TM3, which slides

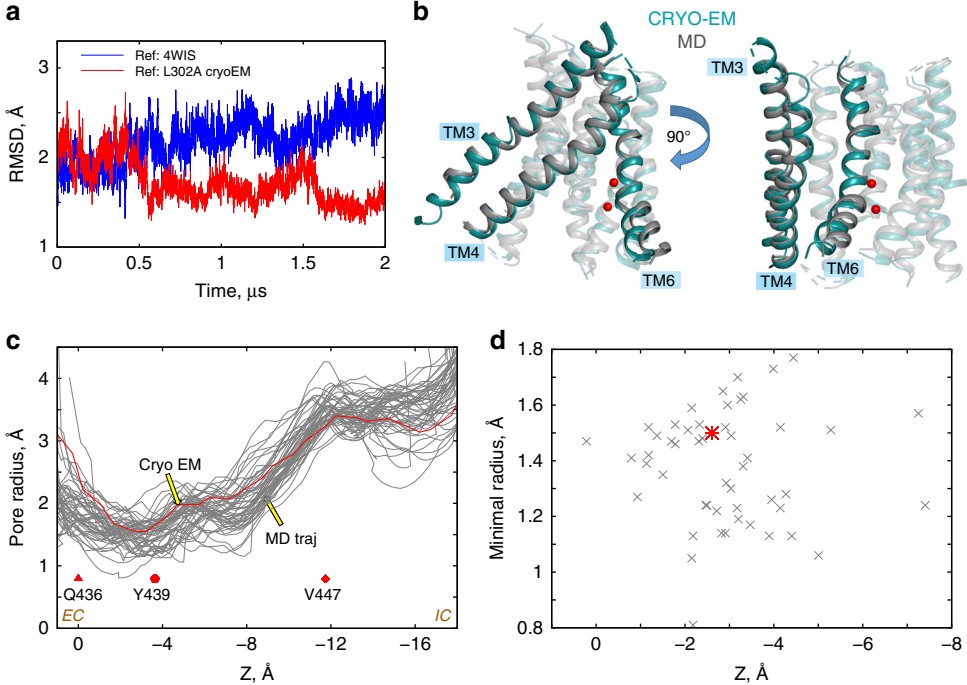

**Fig. 8** Structural similarity between the cryo-EM and computationally predicted models of the L302A nhTMEM16. **a** Time-evolution of the backbone RMSD of the TM helices for one monomer of the L302A nhTMEM16 from a 2 μs-long MD simulation (Supplementary Table 1) with respect to either the cryo-EM model of L302A nhTMEM16 (red), or the X-ray structure of the WT nhTMEM16 from PDBID: 4WIS[7]. **b** Superposition of the TM helices of L302A nhTMEM16 of the cryo-EM model (teal) with the average structure from the last 500 ns timeframe of the MD trajectory (gray). The backbone atom RMSD over all 10 TM helices is 1.2 Å. TM3, TM4, and TM6 helices are opaque and labeled, whereas the rest of the TM-s are shown in transparent. The $Ca^{2+}$ ions from the cryo-EM structure are shown as red sphere. **c** The pore radius as a function of position along an axis perpendicular to the membrane (channel coordinate) for the cryo-EM structure of L302A (red line), and for 50 evenly spaced frames from the last 500 ns of the 2 μs MD simulation of the L302A from (**a**) (gray lines). The EC and IC ends of the pore are indicated, and $Z = 0$ Å corresponds to the location of the $C_\alpha$ atom of residue Q436 (red triangle symbol), the dot marks the position of the $C_\alpha$ atoms of residue of Y439, and the diamond symbol that of V447. The calculations were performed with the program HOLE (http://www.holeprogram.org/). **d** Values of the minimal radius of the pore versus the value of the channel coordinate at which the pore radius is minimal. The calculations are from the HOLE profiles shown in (**c**) (the red colored data point corresponds to the minimum in the cryo-EM structure profile; see red line in (**c**))

upwards and away from TM5 and TM6, and of TM4, which tilts away from TM3 and TM5 and toward TM6 (Fig. 7c), bringing T333 and V337 of TM4 to interact with Y439, T443, and V447 from TM6. This forms a steric barrier that prevents lipid headgroup entry into the groove (Fig. 7e). In the L302A structure, TM3 moves up and away from TM5 and TM6 (Fig. 7c, Supplementary Fig. 12I), disrupting the interactions between E313 and E318 on TM3 and R432 on TM6 (Fig. 7f), resulting in an opened extracellular gate[21] and reorients the side chains of E313 and E318 towards the extracellular vestibule (Supplementary Fig. 14K), while the positively charged R432 side chain is stabilized within the pathway by interactions with several neighboring polar side chains (Fig. 7f). Together, these rearrangements result in a widening of the extracellular vestibule and central region of the pathway, generating a continuous TM pore that is sufficiently wide to accommodate water molecules and ions such as $Na^+$ and $K^+$ (Fig. 7g, h).

Overall, the new cryo-EM structure of nhTMEM16-L302A is in remarkable agreement with the computationally predicted model of this mutant, with a ~1.2 Å backbone RMSD of the TM helices (Fig. 8a, b), while the RMSD with the membrane-exposed nhTMEM16 (PDBID: 4WIS) is twice that (2.7 Å). The pore radii of the L302A groove in the cryo-EM model and in the trajectory frames from the MD simulations are also similar (Fig. 8c, d). Interestingly, in the MD simulations of the L302A mutant, and in the cryo-EM intermediate conformation, the extracellular gate is not fully disengaged. In the most populated ensemble of states the

E318 to R432 distance is large (~13 Å, see panel E318-R432 for microstate a in Supplementary Fig. 9B), but E313-R432 remain locked in ionic interactions (panel E313-R432 for microstate a in Supplementary Fig. 9B). This is consistent with our previous observation[21] that open/closed conformations of the extracellular gate are dynamically sampled.

The continuous TM pore of the L302A structure is isolated from membrane lipids by the interaction of TM4 and TM6 and is the result of the disruption of the hydrophobic lock between TM3 and TM4 (Fig. 7i–h, Supplementary Fig. 15A–D). With a minimum inner diameter > 2.9 Å (Supplementary Fig. 15E), this pore is compatible with the movement of permeant monovalent cations such as $Na^+$ or $K^+$ while sterically precluding passage of large cations, such as $NMDG^+$, which are not conducted by nhTMEM16[24]. In the L302A cryo-EM structure the pore is too narrow to accommodate $Cl^-$ ions, suggesting that further small rearrangements, on the scale of the side chain fluctuations observed in the MD simulations (Fig. 8c), might be required to pass this ion.

**The L302A groove has the property of a non-selective channel.** Our results suggest that the L302A mutant can dynamically sample conformations in which a pore is sufficiently wide to enable ion permeation while remaining occluded from the membrane and preventing the passage of lipids. To evaluate this functional inference, we compared the electrostatic characteristics of the L302A groove region to those of (i)-the $Ca^{2+}$-bound

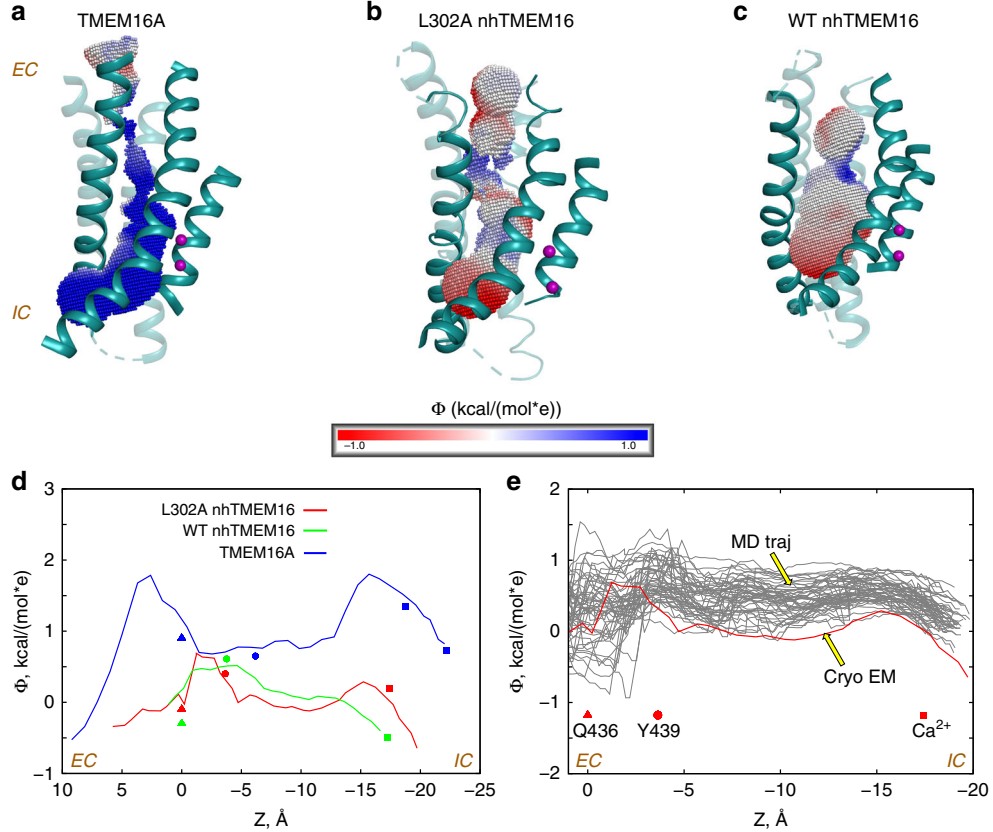

**Fig. 9** Comparative analysis of the electrostatic characteristics of the groove. **a–c** The electrostatic potential in the pore region of various TMEM16 constructs, obtained by solving linear Poisson-Boltzmann equation (see "Methods") are depicted on the surface created by the [−1.0; 1.0] kcal/(mol e) range of values in the groove. The results are shown for: **a** TMEM16A (PDBID 5OYB); **b** the cryo-EM structure of L302A-nhTMEM16; and **c** wild-type nhTMEM16 (PDBID 4WIS). The electrostatic potential in the range of [−1.0; 1.0] kcal/(mol e)) is overlaid on the groove helices (TMs 3, 4, 6, and 7) of the respective structures. The $Ca^{2+}$ ions are shown as purple spheres. The locations of the EC and IC vestibules are marked. **d** The electrostatic potential along the pore axis from the calculations shown in (**a**). The decreasing Z coordinate along the pore axis corresponds to EC → IC direction. The locations of selected relevant residues are marked with different symbols, with the colors specifying the TMEM16 construct as follows: Red = L302A-nhTMEM16; Blue = TMEM16A; and Green = wild-type nhTMEM16. With the respective colors, the symbols represent the following: Triangle = $C_\alpha$ atom of Q436 (Q637 in TMEM16A); Dot = $C_\alpha$ atom of Y439 (I641 in TMEM16A); and Square = $Ca^{2+}$. **e** The electrostatic profile along the pore axis in the cryo-EM structure of L302A-nhTMEM16 (red thick line), and in 50 evenly spaced frames from the last 500 ns of the 2 μs MD simulation of the L302A (gray lines, see also Fig. 8). As in (**b**), the decreasing Z coordinate along the pore axis corresponds to EC → IC direction. The locations of selected relevant residues are marked with symbols following the same code as in (**b**)

wild-type nhTMEM16 scramblase (PDBID 4WIS), and (ii)-the $Ca^{2+}$-bound TMEM16A channel (PDBID 5OYB). The electrostatic potential (EP) profiles along the groove region of each construct was calculated by solving the corresponding linear Poisson-Boltzmann equation ("Methods").

The EP in the groove of the TMEM16A channel is strongly positive, as expected from its known function as a $Cl^-$ channel[9,25] (Fig. 9a). In contrast, the EP in the L302A-nhTMEM16 groove (Fig. 9b) is close to electroneutral, with mildly electronegative patches in the extracellular vestibule near E313, and slightly electropositive ones in the middle section of the groove near the $Ca^{2+}$ binding sites and R505 (Supplementary Fig. 16A). In WT nhTMEM16 (Fig. 9c), the calculated EP profile is close to neutral, as in the L302A groove. The comparative observations from the profiles are supported by the EP values calculated along the pore axis of each construct shown in Fig. 9e where specific residues along the axis are indicated for positional reference. Figure 9f illustrates the fluctuations of EP values along the same axis expected from the dynamics of the protein. Together, these comparisons support the idea that the pathway of the L302A mutation functions as an electrostatically non-selective channel pore.

## Discussion
Our results illuminate the detailed molecular mechanism by which the lipid groove of the nhTMEM16 scramblase can rearrange from a membrane-exposed conformation to an intermediate state that is occluded from the membrane, forming an ion-conductive conformation. These conformational rearrangements are produced by a mechanism that re-positions the mid-portions of TM4 towards TM6 giving rise to a conformation that is unfavorable for lipid scrambling. Following the transition mechanism to this state we uncovered a network of hydrophobic interactions between TM3 and TM4, involving residue L302, which plays a critical role in mediating this conformational change. Mutating L302 to alanine stabilizes a $Ca^{2+}$-bound conformation of the nhTMEM16 groove with significantly impaired scrambling activity but retaining ion channel function. The mechanism of conformational transitions described here suggests how transitions can occur among groove conformations in the TMEM16 channels and scramblases, which can enable the translocation of substrates as different as ions and lipids.

The state described here is structurally similar in the groove region to that seen in the intermediate conformation of $Ca^{2+}$-bound nhTMEM16[18], but with a distinct structural feature: the

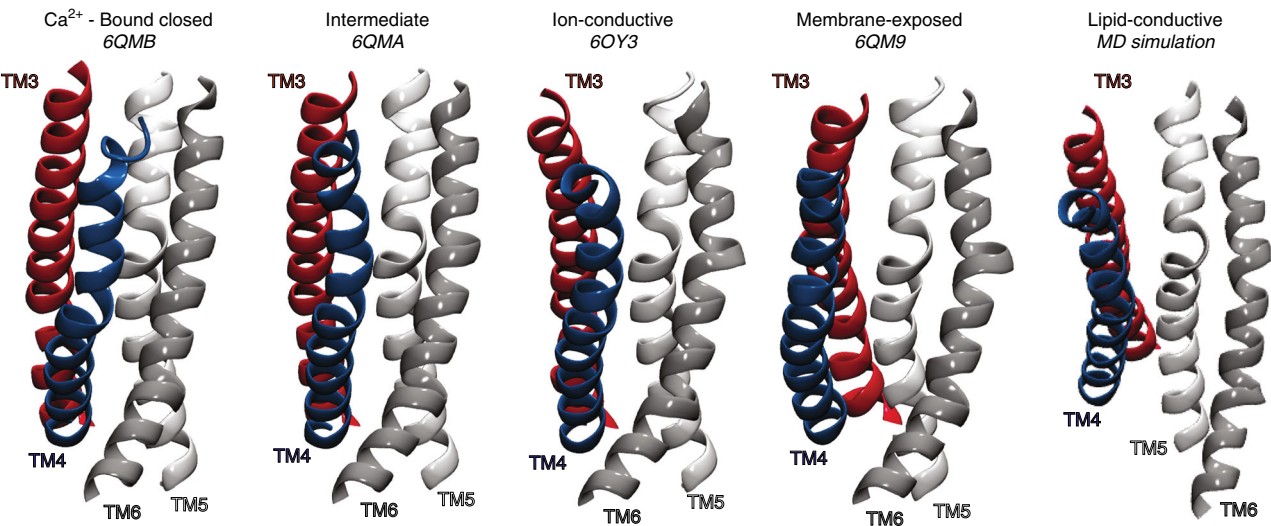

**Fig. 10** Structural and functional characteristics of the known conformations of the groove of $Ca^{2+}$-bound nhTMEM16 scramblase. The groove conformations shown are (from left to right): $Ca^{2+}$-bound closed (PDBID 6QMB[18]); intermediate (PDBID 6QMA[18]); ion-conductive (PDBID 6OY3, this study); membrane-exposed (PDBID 6QM9[18]); and lipid-conductive (obtained from MD simulations[21]). In all structures, TMs 3–6 are colored in red, blue, white, and silver, respectively, and are labeled

EC gate of the scramblase[21] is open because of a pronounced repositioning of TM3 away from TM6 on the EC side of the groove. Occurrence of this $Ca^{2+}$-bound intermediate in the nhTMEM16 scramblase relates to lipid-mediated disruption of hydrophobic interactions between TM3 and TM4. We identify the main residues involved in constraining the two TM segments in the membrane-exposed and lipid-conductive conformations of the groove, and in controlling the transition to the intermediate conformation. In our MD simulations we observed that the evolution from the membrane-exposed state to the intermediate one is facilitated by insertion in the groove of a lipid tail that interacts simultaneously with V337 on TM4 and V447 on TM6, bringing them close together. Once this proximity is established in the presence of the inserted lipid tail, however, it persists after the inserted tail has diffused out of the groove. While we have not observed an event of groove closure not preceded by a lipid tail insertion, it is possible that the occlusion can occur also without lipid participation. Our results show that the conformation of the groove described here can still conduct ions but not lipids, thus we refer to this conformation as ion conductive. Our data do not rule out that in nhTMEM16, ions are translocated as well through the lipid-conductive conformation, as previously observed in MD simulations[14].

An essential element in the mechanism of transformations that modulate the groove in nhTMEM16 is the reorientation flexibility of TM4. We found that switching between the open and intermediate states of the groove involves a transition between relatively straight and bent conformations of TM4. In our simulations of WT nhTMEM16 the occurrence of the intermediate state was rare, but the nature of the rearrangements leading to stabilization of this state allowed us to identify mutations that preferentially stabilize this intermediate state. Indeed, three mutations designed to disrupt the TM3/TM4 interaction that maintain the membrane-exposed state with a lipid-conductive groove conformations, severely impair scrambling activity in vitro. Moreover, the prediction from the simulation analysis that L302A mutation would preferentially stabilize the protein in the intermediate state was verified directly by solving the structure of the L302A-nhTMEM16 construct reconstituted into nanodiscs using cryo-EM. This structure revealed an intermediate state that resembles very closely the one predicted by the

simulations. In contrast to the WT nhTMEM16[18], several rounds of 3D classification of cryo-EM data for L302A did not reveal additional groove conformations, suggesting that the mutation stabilizes this conformation.

Our findings suggest that the membrane-exposed conformation of nhTMEM16 is stabilized by hydrophobic interactions between TM3 (L302) and TM4 (I343A, L347A), while hydrophobic contacts between TM4 (V337) and TM6 (V447) favor the channel-like state. Indeed, our recent finding that the V337W mutant of nhTMEM16 has increased channel activity in the absence of $Ca^{2+}$[21], is consistent with the idea that these residues regulate the conformation of the nhTMEM16 scramblase. Further, the channel-only TMEM16A could be converted into a scramblase by the V543T/S mutations, one turn above the residue corresponding to V337 in nhTMEM16[14]. Together, these observations underscore the importance of the hydrophobicity of the central region of TM4 for TMEM16 channel function. In line with past proposals[14], we suggest that mutations in this region may shift the conformational equilibrium between the channel-like (intermediate) and open-groove states, either by altering the interactions between TM4 and TM6[14,21] or by decoupling TM4 from TM3, as seen here to result from the L302A mutation.

That the intermediate state in nhTMEM16 is not permissive to lipid head-group penetration but can still support ionic conductance as shown here, suggests that the groove of nhTMEM16 can mediate transport of different substrates by adopting distinct conformations (Fig. 10). The minimum pore diameter of nhTMEM16 L302A is ~2.9 Å, wide enough for $Na^+$ and $K^+$ ions, and larger than those measured in the $Ca^{2+}$-bound or $Ca^{2+}$-free closed conformations of other scramblases[17–19] as well as in the TMEM16A channel[9,10]. While in the L302A cryo-EM structure the pore is too narrow for $Cl^-$ permeation, our MD simulations suggest that subtle rearrangements of the pore-lining side chains would suffice for $Cl^-$ transport.

The structure of the L302A mutant was determined in nanodiscs formed from a 3:1 mixture of POPE:POPG lipids, in which WT nhTMEM16 does not exhibit non-selective channel activity[24]. It is possible that specific interactions of the lipid head-groups with residues in the TM4 and TM6 helices, and/or changes in overall membrane properties, might favor the small rearrangements necessary to allow $Cl^-$ passage, so that the pore

becomes more, or less, conductive to ions. Indeed, interactions between lipid headgroups and residues lining the lipid groove are important to facilitate opening of the EC gate to enable the adoption of a lipid-conductive conformation of the pathway[21]. We do observe groove closure events in POPE/POPG and in POPC membranes in our MD simulations, but the sampling of these events is insufficient for a statistically significant quantification of the likelihood for closure in these different lipid compositions. Notably, the membrane compositions that inhibit channel activity of nhTMEM16 and afTMEM16, have high melting temperatures, close to room temperature, while those that are permissive for channel activity have lower melting temperatures[24,26]. This raises the possibility that the bilayer phase properties might affect the transition probability to, or the stability of, the ion conductive conformation of the nhTMEM16 scramblase groove. Indeed, the recent reports suggesting that reconstitution in nanodiscs lowers the melting temperature of lipids[27] would be consistent with the idea that the structure of L302A nhTMEM16 in POPE:POPG nanodiscs reflects an ion conductive conformation. Further experiments are needed to test this possibility.

It is noteworthy that in our longest MD trajectory (15 μs-long simulation of the WT protein) we observed partial $Cl^-$ ion permeation through the intermediate state from the IC vestibule all the way to the EC milieu. The penetrating $Cl^-$ ion is attracted to residue R505 in TM7 and then continues its path towards the EC vestibule up to the level of Y439 (Supplementary Fig. 16B). These observations agree with our findings of a slightly electropositive region next to R505 in the L302A pore (Supplementary Fig. 16A). In view of our previous description of the EC gating mechanism for lipid flipping[21] it is tempting to speculate that complete ion translocation events would require dissolution of the EC gates, as seen in our cryo-EM structure. Further experiments are needed to test this hypothesis as well as the role of membrane voltage and ionic gradients in ion permeation and pore gating in the TMEM16s.

The remarkable ability of the TMEM16 proteins to transport dissimilar substrates, such as ions and lipids, despite sharing highly conserved structural architectures has constituted a longstanding mechanistic puzzle[5,6]. Our present work, together with the recent structures of several TMEM16 proteins[7,9,10,17–20,22], are beginning to reveal the spectrum of structural conformations adopted by the groove of the TMEM16 scramblases and to identify the functional properties of these states (Fig. 10). On this basis we propose a mechanistic framework where, after binding of the regulatory $Ca^{2+}$ ions, transitions can occur away from a closed conformation of the groove (Fig. 10, $Ca^{2+}$-bound closed)—in which TM4 interacts with TM6 to prevent lipid access and is situated close to TM5 preventing ion movement—so that rearrangements of TM3 and TM4 generate an intermediate (Fig. 10, intermediate), or an ion-conductive conformation (Fig. 10, ion-conductive). In the latter, the persisting engagement of TM4 and TM6 prevents lipid access, but ion permeation is allowed. Upon disruption of the interactions between TM4 and TM6, the pathway opens to reveal its hydrophilic interior to the hydrocarbon core of the membrane (Fig. 10, membrane exposed). In this conformation the bilayer is primed for lipid penetration[19], but the further opening of the extracellular and central gates of the pathway is necessary to allow the rapid trans-bilayer movement of lipids (Fig. 10, Lipid-conductive). Within this framework, the emergence of channel-only TMEM16 homologues would require a change in the conformational preference of the TMEM16 molecule, with destabilization of the open and lipid-conductive conformations of the groove. This hypothesis is consistent with our present results, as well as with the findings that point mutations can transform the TMEM16A channel into a scramblase and the TMEM16F scramblase into a channel[14,21,28].

## Methods

**Molecular constructs for MD simulations.** The computational analysis presented in this work is based on extensive all-atom MD simulations of the wild-type (WT) nhTMEM16 protein (the TMEM16 homologue from Nectria haematococca)[7] and of the experimentally characterized nhTMEM16 mutants A385W, L302A, and L302W (see Supplementary Table 1 for a complete listing). The ensemble of MD trajectories analyzed here includes those that have been already described in our recent study[21] as well as those resulting from the new simulations described below.

The full-length model of the WT nhTMEM16 used in all the simulations was based on the X-ray structure from PDBID: 4WIS[7] in which several fragments were missing (residue segments 1–18, 130–140, 465–482, 586–593, 657–659, 685–691, and 720–735). For the computed model, these segments were reconstructed using modeler 9v1[29] to complete the protein structure. All the mutations were introduced into the full-length protein with the BuildModel algorithm available in the FoldX suite[30].

The WT and the mutant nhTMEM16 constructs were simulated in lipid bilayers. Their spatial arrangement was optimized using the Orientations of Proteins in Membranes (OPM) database[31] and inputted into the Membrane Builder module on CHARMM-GUI web-server[32] in order to assemble protein-membrane systems. All the protein models were embedded in 1500-lipid size membranes containing 3:1 mixture of POPE (1-palmitoyl-2-oleoyl-sn-glycero-3-phosphoethanolamine) and POPG (1-palmitoyl-2-oleoyl-sn-glycero-3-phospho-(1′-rac-glycerol)) lipids. For WT nhTMEM16, additional simulations were carried out in POPC (1-palmitoyl-2-oleoyl-sn-glycero-3-phosphocholine) and 3:1 POPE/POPG bilayers composed of 680 lipids. The protein-membrane complexes were solvated in 0.15 M $K^+Cl^-$ explicit water solution to achieve electroneutrality.

**MD simulations protocols and force-fields.** Several sets of MD simulations analyzed in the present work have been described in detail in Ref. [21]. and are identified by a * symbol in Supplementary Table 1. These include simulations on the Anton2 special-purpose supercomputer machine[33] of the wild-type nhTMEM16 (WT1, WT2 in Supplementary Table 1) and of the A385W mutant, all in 1500-lipid size 3:1 POPE/POPG membrane, as well as the ensemble MD simulations of the wild-type construct in 680-lipid size POPC membrane (WT^ensemble in Supplementary Table 1) performed with the ACEMD software[34]. These trajectories are complemented here with a new set of microsecond-scale MD simulations of the wild type and the two mutant nhTMEM16 constructs —L302A and L302W—run on the Anton2 machine. The new simulation data include two independent trajectories of the WT nhTMEM16 (WT5, WT6 in Supplementary Table 1), 15 and 10 μs in length, in 680-lipid size POPC and 3:1 POPE/POPG bilayers, respectively. In addition, two 1 μs long independent MD trajectories of the WT nhTMEM16 were run in 1500-lipid size POPC bilayer (WT3, WT4 in Supplementary Table 1). The L302A and L302W mutants were simulated in 1500-lipid size 3:1 POPE/POPG membranes and were run in two replicates (in total, 1 and 2 μs trajectories were accumulated for the L302A construct, and 3 and 4 μs trajectories for the L302W).

Like the previous simulations[21], the protocols for WT nhTMEM16 embedded in POPC and 3:1 POPE/POPG bilayers, as well as the L302A and L302W systems, included first the multi-step equilibration protocol established previously[35], using NAMD software version 2.10[36]. During this stage, the backbone of the protein was first fixed and then harmonically restrained with the constraints gradually released in three steps of 1 ns each by changing the force constants from 1, to 0.5, and 0.1 kcal/(mol Å²), respectively. This phase was followed by unbiased MD simulations performed with a 2 fs integration time-step, under the NPT ensemble (at $T = 310$ K) and with semi-isotropic pressure coupling, using the Particle-Mesh-Ewald (PME) method for electrostatics[37] and the Nose-Hoover Langevin piston[38] to control the target 1 atm pressure, with Langevin piston period and decay parameters set to 100 fs and 50 fs, respectively. For the WT systems in 680-lipid size POPC and 3:1 POPE/POPG membranes (WT5-WT6), the unbiased MD phase was run for ~380 ns. For the L302A and L302W mutant constructs in POPE:POPG bilayer, as well as for the WT system in the 1500-lipid size POPC membrane (WT3-WT4) the unbiased MD phase was run in two statistically independent replicates (with new sets of velocities generated by random seed) for 22 ns and 104 ns for the L302A system, for 20 ns and 168 ns for the L302W system, and for 24 ns and 27 ns for the WT system.

After this initial phase, all the molecular systems were subjected to microsecond-scale MD simulations on Anton2 (the timescales are indicated in Supplementary Table 1). These production runs were carried out in the NPT ensemble under semi-isotropic pressure coupling conditions (using the Multigrator scheme that employs the Martyna-Tuckerman-Klein (MTK) barostat[39] and the Nosé-Hoover thermostat[40]), at 310 K temperature, with 2.5 fs time-step, and using PME for electrostatic interactions. All the simulations implemented the CHARMM36 force-field parameters for proteins[41], lipids[42,43], and ions[44].

**Dimensionality reduction with the tICA approach.** To facilitate comparative analysis of structural characteristics of the WT and the mutant (L302A, L302W) systems, we performed dimensionality reduction using tICA (time-lagged independent component analysis)[45] as described before[21,35,46,47]. Briefly, the tICA approach utilizes the MD simulation trajectories to construct two covariance matrices, one being a time-lagged covariance matrix (TLCM): $\mathbf{C_{TL}}(\tau) = <\mathbf{X}(t)\mathbf{X}^T$

$(t + \tau) >$ and the other the usual covariance matrix $\mathbf{C} = < \mathbf{X}(t)\mathbf{X}^T(t) >$, where $\mathbf{X}(t)$ is the data vector at time t, τ is the lag-time of the TLCM, and the symbol <…> denotes the time average. The slowest reaction coordinates of the system are then identified by solving the generalized eigenvalue problem: $\mathbf{C_{TL}V} = \mathbf{CV\Lambda}$, where $\mathbf{\Lambda}$ and $\mathbf{V}$ are the eigenvalue and eigenvector matrices, respectively. The eigenvectors corresponding to the largest eigenvalues define the slowest reaction coordinates. These reaction coordinates depend on the choice of data vector $\mathbf{X}$. To define the tICA space we used the following five dynamic variables extracted from the analysis of the MD trajectories (see also Supplementary Table 2 and Results for more details): (1)-the minimal distance between T333 and Y439 (i.e., the distance between the closest pair of atoms on the two residues); (2)-the minimal distance between Y439 and R432; (3–4) distance between E313 and R432 and between E318 and R432 (defined as distance between carbonyl oxygen of Glu and sidechain nitrogen of Arg); and (5)-the $C_\alpha$-$C_\alpha$ distance between V337 and V447 residues.

**Definition of the hydrophilic groove volume.** To quantify the hydration of the groove and of its segments we defined several volumes and counted numbers of waters in these volumes as done previously[21]. A water molecule was considered to belong to the groove if any of its atoms was within 3 Å of the sidechains of the following residues lining the groove area: 302, 306, 310, 313, 333, 336, 337, 340, 341, 344, 348, 352, 367, 370, 371, 374, 377, 378, 381, 382, 385, 432, 436, 439, 440, 444, 447, 451, 455, 499, 501, 505, 509, and 513. A water molecule was considered to belong to the extracellular vestibule of the groove if the z-coordinate (along membrane normal) of any atom of the water molecule was found between the x/y (membrane) planes defined by the $C_\alpha$ atoms of residues T381 and Q436.

Similarly, in order to quantify the penetration of lipid head groups or acyl chains into the groove we counted the number of non-hydrogen atoms on the lipid head group, or the tail, that were within 5 Å of the sidechains of the following residues lining the interior of the groove area: 377, 378, 381, 382, 385, 501, 505, 509, and 513. These counts were then normalized by the total number of these non-hydrogen atoms (on the lipid head group or tail, respectively). For these definitions, the lipid head group was considered to extend down to the C21-O21 and C31-O32 carbonyls in the lipid backbone (in CHARMM force-field nomenclature), and the lipid tail represents the rest of the hydrocarbon chains.

**Calculation of the EP.** The EP was calculated by solving the linearized Poisson-Boltzmann equation in CHARMM[48,49]. The protein was assigned a dielectric constant of $\varepsilon_p = 2$. Its TM region was embedded in a membrane slab, which included a 26-Å-thick hydrophobic core of the bilayer (with $\varepsilon_{core} = 2$ dielectric constant), surrounded by two 8-Å-thick regions, representing the lipid headgroups ($\varepsilon_{head} = 30$ dielectric was set in this region). The membrane slab contains a cylindrical hole (with the dielectric of $\varepsilon_W = 80$) with a radius of 18 Å centered around the groove of the protomer. The system is surrounded by solvent ($\varepsilon_W = 80$) containing 150 mM of monovalent mobile ions. The cylindrical hole is accessible to the ions, but ions are not permitted into the membrane region.

The axis and the radius of the hydrophilic pore of the TMEM16 constructs studied here were calculated with the program HOLE[50] (http://www.holeprogram.org/), and the pore region was determined by including all points lying within the space defined by the radius of the axis, but distant from the protein by more than 2.5 Å.

**Protein expression and purification.** Wild-type and mutant nhTMEM16 for functional studies were expressed from a construct with a C-terminal Myc-streptavidin-binding peptide (SBP)-tag and wild-type and mutant nhTMEM16 for structural studies were expressed using a construct with an N-terminal GFP-His$_{10}$[7]. Expression and purification from both constructs were carried out essentially as described[7,21]. S. cerevisiae cell (FGY217, from Dr. Drew, University of Stockholm)[51] were transformed with WT and mutant constructs, grown to an O.D. of 0.8 and protein expression was induced by the addition of 2% galactose for 40 h at 25 °C. For the C-terminal Myc-SBP construct, cells were resuspended in lysis buffer (150 mM NaCl, 50 mM HEPES, pH 7.6) containing protease inhibitor cocktail and lysed with an EmulsiFlex-C3 homogenizer at above 25,000 psi. Membrane proteins were extracted by supplementing the lysis buffer with 2% n-dodecyl-β-D-maltopyranoside (DDM, Ana- trace), and incubated for 1.5 h at 4 °C. Proteins were purified using Streptavidin Plus UltraLink Resin (Pierce) followed by gel filtration chromatography with buffer A (150 mM NaCl, 5 mM HEPES pH 7.6, 0.025% DDM) by using a Superdex 200 column (GE Healthcare). For the N-terminal GFP-His$_{10}$ construct, buffers were supplemented with CaCl$_2$ (0.5 mM for lysis and extraction, 5 mM for affinity chromatography, and 3 mM for size exclusion) and extraction was carried out with 1% DDM. Protein was purified using NiNTA resin equilibrated with 15 mM Imidazole and eluted using 400 mM Imidazole. The N- terminal GFP-His$_{10}$ tag was cleaved overnight with 3C protease before gel filtration chromatography.

**Liposome reconstitution and lipid scrambling assay.** Liposomes were prepared as described from a 2.25:0.75:1 mixture of 1-palmitoyl-2-oleoyl-sn- glycero-3-phosphoethanolamine (POPE): 1-palmitoyl-2-oleoyl-sn-glycero-3-phos- pho-(1′-rac-glycerol) (POPG) and L-α-Phosphatidylcholine (Egg, Chicken-60%) including 0.4% w/w 1-myristoyl- 2-{6-[(7-nitro-2-1,3-benzoxadiazol-4-yl)amino]hexanoyl}-sn-glycero-3- phosphoethanolamine (NBD-PE). All lipids were purchased from

Avanti Polar Lipids[21,26]. Briefly, lipids were dried under N$_2$, washed with pentane and resuspended at 20 mg ml$^{-1}$ in buffer B (150 mM KCl, 50 mM HEPES pH 7.4) with 35 mM 3-[(3-cholamidopropyl)dimethylammonio]-1- propanesulfonate (CHAPS). nhTMEM16 was added at 5 μg protein/mg lipids and detergent was removed using four changes of 150 mg ml$^{-1}$ Bio-Beads SM-2 (Bio-Rad) with rotation at 4 °C. Calcium or EGTA were introduced using sonicate, freeze, and thaw cycles. Liposomes were extruded through a 400 nm membrane and 20 μl were added to a final volume of 2 mL of buffer B + 0.5 mM Ca(NO$_3$)$_2$ or 2 mM EGTA. The fluorescence intensity of the NBD (excitation-470 nm emission-530 nm) was monitored over time with mixing in a PTI spectrophotometer and after 100 s sodium dithionite was added at a final concentration of 40 mM. Data were collected using the FelixGX 4.1.0 software at a sampling rate of 3 Hz.

**Quantification of scrambling activity.** Quantification of the scrambling rate constants by nhTMEM16 was determined as recently described[21,52]. Briefly, the fluorescence time course was fit to the following equation

$$F_{tot}(t) = f_0\left(L_i^{PF} + (1 - L_i^{PF})e^{-\gamma t}\right) + \frac{(1 - f_0)}{D(\alpha + \beta)}$$
$$\{\alpha(\lambda_2 + \gamma)(\lambda_1 + \alpha + \beta)e^{\lambda_1 t} + \lambda_1\beta(\lambda_2 + \alpha + \beta + \gamma)e^{\lambda_2 t}\} \quad (1)$$

Where $F_{tot}(t)$ is the total fluorescence at time t, $L_i^{PF}$ is the fraction of NBD-labeled lipids in the inner leaflet of protein-free liposomes, $\gamma = \gamma'[D]$ where $\gamma'$ is the second order rate constant of dithionite reduction, [D] is the dithionite concentration, $f_0$ is the fraction of protein-free liposomes in the sample, α and β are respectively the forward and backward scrambling rate constants and

$$\lambda_1 = -\frac{(\alpha + \beta + \gamma) - \sqrt{(\alpha + \beta + \gamma)^2 - 4\alpha\gamma}}{2},$$
$$\lambda_2 = -\frac{(\alpha + \beta + \gamma) + \sqrt{(\alpha + \beta + \gamma)^2 - 4\alpha\gamma}}{2}, \quad (2)$$
$$D = (\lambda_1 + \alpha)(\lambda_2 + \beta + \gamma) - \alpha\beta$$

The free parameters of the fit are $f_0$, α and β while $L_i^{PF}$ and γ are experimentally determined from experiments on protein-free liposomes. In protein-free vesicles a very slow fluorescence decay is visible, likely reflecting a slow leakage of dithionite into the vesicles or the spontaneous flipping of the NBD-labeled lipids. A linear fit was used to estimate the rate of this process was estimated to be L = (5.4 ± 1.6)×10$^{-5}$ s$^{-1}$ (n > 160). For WT nhTMEM16 and most mutants the leak is >2 orders of magnitude smaller than the rate constant of protein-mediated scrambling and therefore is negligible. All conditions were tested side by side with a control preparation in standard conditions. In some rare cases this control sample behaved anomalously, judged by scrambling fit parameters outside three times the standard deviation of the mean for the WT. In these cases the whole batch of experiments was disregarded.

**Flux assay.** Cl$^-$ flux assay was conducted as described previously[21,24,52]. Liposomes with Ca$^{2+}$ or EGTA introduced were equilibrated in external buffer with low KCl (1 mM KCl, 300 mM Na-glutamate, 50 mM HEPES, pH 7.4) by spinning through a Sephadex G50 column (Sigma-Aldrich) pre-equilibrated in external buffer. To complete the experiment, 0.2 mL of the flow through from the G50 column was added to 1.8 ml of external solution and the total Cl$^-$ content of the liposomes was measured using an Ag:AgCl electrode after disruption of the vesicle by addition of 40 μL of 1.5 M n-octyl-β-D- glucopyranoside (Anatrace). The fraction of liposomes containing at least one active nhTMEM16 ion channel, A, was quantified as follows:

$$A = 100 * \left(1 - \frac{\Delta Cl}{\Delta Cl_{PF}}\right) \quad (3)$$

where ΔCl is the change in [Cl$^-$] recorded upon detergent addition and $\Delta Cl_{PF}$ is the Cl$^-$ content of protein-free liposomes prepared in the same lipid composition on the same day.

**MSP1E3 purification and nanodisc reconstitution.** MSP1E3 was expressed and purified as described[53]. Briefly, MSP1E3 in a pET vector (Addgene #20064) was transformed into the BL21-Gold (DE3) strain (Stratagene). Transformed cells were grown in LB media supplemented with Kanamycin (50 mg l$^{-1}$) to an OD$_{600}$ of 0.8 and expression was induced with 1 mM IPTG for 3 h. Cells were harvested and resuspended in buffer C (40 mM Tris-HCl pH 78.0, 300 mM NaCl) supplemented with 1% Triton X-100, 5 μg ml$^{-1}$ leupeptin, 2 μg ml$^{-1}$ pepstatin, 100 μM phenylmethane sulphonylfluoride and protease inhibitor cocktail tablets (Roche). Cells were lysed by sonication and the lysate was cleared by centrifugation at 30,000 g for 45 min at 4 °C. The lysate was incubated with Ni-NTA agarose resin for one hour at 4 °C followed by sequential washes with: buffer C + 1% triton-100, buffer C + 50 mM sodium cholate + 20 mM imidazole and buffer C + 50 mM imidazole. The protein was eluted with buffer C + 400 mM imidazole, desalted using a PD-10 desalting column (GE life science) equilibrated with buffer D (150 mM KCl, 50 mM Tris pH 8.0) supplemented with 0.5 mM EDTA. The final protein was concentrated

to ~8 mg ml$^{-1}$ (~250 μM) using a 30 kDa molecular weight cut off concentrator (Amicon Ultra, Millipore), flash frozen and stored at −80 °C.

Reconstitution of nhTMEM16 in nanodiscs was carried out as described for afTMEM16[19], briefly, 3POPE:1POPG lipids in chloroform (Avanti) were dried under N$_2$, washed with pentane and resuspended in buffer D and 40 mM sodium cholate (Anatrace) at a final concentration of 20 mM. Molar ratios of 1:0.8:40 MSP1E3:nhTMEM16:lipids were mixed at a final lipid concentration of 7 mM and incubated at room temperature for 20 min. Detergent was removed via incubation with 200 mg mL$^{-1}$ Bio-Beads SM-2 (Bio-Rad) at room temperature with agitation for one hour, three times, with a change of the biobeads after each hour (Bio-Beads SM2 at a concentration of 200 mg mL$^{-1}$). The reconstitution mixture was purified using a Superose6 Increase 10/300 GL column (GE Lifesciences) pre-equilibrated with buffer D plus 0.5 mM CaCl$_2$ and the peak corresponding to nhTMEM16-containing nanodiscs was collected for cryo electron microscopy analysis.

**Electron microscopy data collection.** 3.5 μL of nhTMEM16-containing nanodiscs (6.5 mg mL$^{-1}$) supplemented with 3 mM Fos-Choline-8-Fluorinated (Anatrace) was applied to a glow-discharged UltrAuFoil R1.2/1.3 300-mesh gold grid (Quantifoil) and incubated for one minute under 100% humidity at 15 °C. Following incubation, grids were blotted for 2 s and plunge frozen in liquid ethane using a Vitrobot Mark IV (FEI). Micrographs were acquired using Leginon[54] on a Titan Krios microscope (FEI) operated at 300 kV with a K2 Summit direct electron detector (Gatan), using a slit width of 20 eV on a GIFQuantum energy filter and a Cs corrector with a calibrated pixel size of 1.0961 Å /pixel. A total dose of 70.7 e$^-$/Å$^2$ distributed over 50 frames (1.41 e$^-$/Å$^2$/frame) was used with an exposure time of 10 s (200 ms/frame) and defocus range of −1.5 to −2.5 μm.

**Image processing.** Micrographs were manually inspected and 2696 were included for analysis carried out using Relion 3. Motion correction was carried out using the Relion 3 implementation of MotionCorr2[55,56] and contrast transfer function (CTF) estimation was performed using CTFFIND4[57] via Relion[56]. Manual particle picking was used to pick ~2000 particles, which were classified in 2D and used as templates for automated particle picking. 1,024,244 particles were extracted using a box size of 275 Å with 2xbinning and subjected to two rounds of 2D classification. 333,894 particles with structural features resembling the nhTMEM16-nanodisc complex were selected and subjected to 3D classification without symmetry. 225,926 particles from well-defined 3D classes were selected for 3D refinement without imposed symmetry. The resulting 4.5 Å 3D reconstruction showed twofold symmetry, which was enforced during the remainder of the analysis. Several rounds of 3D classification, refinement, CTF refinement, and Bayesian polishing were carried out leading to a final reconstruction with 47,243 particles and a masked resolution of 4.0 Å. 3D classes were inspected after each round for additional conformations, but none were observed (Supplementary Fig. 13). All removed particles appeared to be in the same conformation, yet at lower resolution. The resolution of refinements with all initially selected 225,926 particles never surpassed 4.3 Å, indicating the excluded particles were indeed at lower resolution. The final resolution of all maps was determined by applying a soft mask around the protein and the gold-standard Fourier shell correlation (FSC) = 0.143 criterion using Relion Post Proccessing (Supplementary Fig. 13E). BlocRes from the Bsoft program was used to estimate the local resolution for all final maps[58,59] (Supplementary Fig. 13F).

**Model building and refinement.** The experimental EM map was of high quality and allowed for the building of an atomic model containing the TM domain, most of the cytosolic region and connecting loops (Supplementary Fig. 14; Supplementary Table 4). The crystal structure of wild-type nhTMEM16 (PDBID 4WIS) was used as a starting model for the TM domain and the EM structure of wild-type nhTMEM16 in detergent (6QM5) was used as the starting model for the cytosolic domain. Several rounds of PHENIX real space refinement[10,60,61] including morphing and simulated annealing every macrocycle were used to fit the wild-type structure into the experimental map. After fitting, TM3 and 4 were manually built as these differed most from the wild-type structure. The final model contains residues 16–97, 101–128, 141–267, 276–319, 324–403, 427–468, 478–584, 594–652, and 695–719 and the following residues were truncated due to missing side chain density: E32, E37, R82, D87, K142, N317, K325, Y327, L328, K467, E480, E497, K598, and K643. To assist building in areas with lower resolution (i.e., TM3 and TM4) the unfiltered 3D reconstruction directly from 3D refinement was used. The model was improved iteratively by real space refinement in PHENIX imposing crystallographic symmetry and secondary structure restraints followed by manual inspection and removal of outliers. As mentioned above, the density in the regions of TM3 and TM4 was less well-defined and of lower resolution than remainder of the structure and the positions of side chains may be less accurate in this area.

To validate the refinement, the FSC between the refined model and the final map was calculated (FSCsum). To evaluate for over-fitting, random shifts of up to 0.3 Å were introduced in the final model and the modified model was refined using PHENIX against one of the two unfiltered half maps. The FSC between this modified-refined model and the half map used in refinement (FSCwork) was determined and compared to the FSC between the modified-refined model and the other half map (FSCfree) which was not used in validation or refinement. The

similarity in these curves indicates that the model was not over-fit (Supplementary Fig. 12G). The quality of all three models was assessed using MolProbity[62] and EMRinger[63], both of which indicate that the models are of high quality (Supplementary Fig. 12G).

**Difference map calculation.** We used omit density to assign the placement the Ca$^{2+}$ ions which were calculated using the phenix.real_space_diff_map function in PHENIX[10,60,61]. Briefly, completed models without the ligand in question were used to generate a theoretical density map which was subtracted from the experimental density map. In the subtracted map, areas of ligand density appeared as positive density[64].

**Reporting summary.** Further information on research design is available in the Nature Research Reporting Summary linked to this article.

## Data availability
Data supporting the findings of this manuscript are available from the corresponding authors upon reasonable request. A reporting summary for this Article is available as a Supplementary Information file. The source data underlying Fig. 6 and Supplementary Fig. 11 are provided as a Source Data file. Final masked and unmasked cryo-EM maps have been deposited in the EMDB database under the accession codes EMD-20221. The map with C1 symmetry which was used to compare the membrane environment and an unfiltered map that was used to aid model building were also included. Atomic coordinates have been deposited in the PDB database under the accession code 6OY3.

## Code availability
For the molecular constructs used in computational experiments, we utilized Modeller version 9v1 and VMD version 1.9.1. To carry out atomistic molecular dynamics simulations, we used NAMD version 2.11, ACEMD version 2, and Anton2. Computational analysis was carried out using a combination of VMD version 1.9.1, python scripts, and in-house scripts based on C code (freely available on GitHub, https://github.com/khelgeo/L302A_Nat_Comm). Computational data used to arrive at the conclusions presented in the manuscript are available upon reasonable request.

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

## Acknowledgements

The authors thank members of the Accardi and Weinstein labs for helpful discussions. We gratefully acknowledge Asghar Razavi for his expert help with the tICA analysis. This work was supported by NIH Grant R01GM106717 (to A.A and H.W.), an Irma T. Hirschl/Monique Weill-Caulier Scholar Award (to A.A.), by the KBRI Basic Research Program funded by the Ministry of Science and ICT (19-BR-01-02) and National Research Foundation of Korea (NRF) grant funded by the Korea government(MSIT) (2019R1C1C1002699 to B.-C. L.). H.W. and G.K. gratefully acknowledge support from the 1923 Fund. The computational work was performed using the following resources: the Extreme Science and Engineering Discovery Environment (XSEDE, account TG-MCB120008), which is supported by National Science Foundation grant number ACI-1053575; the computational resources of the David A. Cofrin Center for Biomedical Information in the HRH Prince Alwaleed Bin Talal Bin Abdulaziz Alsaud Institute for Computational Biomedicine at Weill Cornell Medical College; alsaud Anton 2 super-computer provided by the Pittsburgh Supercomputing Center (PSC) through Grant *R01GM116961* from the National Institutes of Health. The Anton 2 machine at PSC was generously made available by D.E. Shaw Research. M.E.F. is the recipient of a Weill Cornell Medicine Margaret & Herman Sokol Fellowship. All EM data collection and screening were performed at the Simons Electron Microscopy Center and National Resource for Automated Molecular Microscopy located at the New York Structural Biology Center, supported by grants from the Simons Foundation (349247), NYSTAR, and the NIH National Institute of General Medical Sciences (GM103310). Initial negative stain screening was performed at the Weill Cornell Microscopy and Image Analysis Core Facility, with the help of L. Cohen-Gould.

## Author contributions

G.K., M.F., B.-C.L., A.A., and H.W. designed the experiments, M.F. and B.-C.L. performed experiments, G.K. performed the molecular dynamics simulations, G.K., M.F., X.C., B.-C.L., A.A., and H.W. analyzed the data, G.K., M.F., X.C., A.A., and H.W. wrote the paper. All authors edited the manuscript.

## Competing interests

The authors declare no competing interests.
