## [Peer Review File · Nature Communications]

Reviewers' Comments:

Reviewer #1:

Remarks to the Author:

TMEM16 family of membrane proteins displays a remarkable functional dichotomy, with some members working as selective anion channels, and others primarily as lipid scramblases with reported non-selective channel activity. Structural work conducted on TMEM16 channels and scramblases allowed to identify architectural differences between the two functional branches of the family – while the scramblases upon activation expose a hydrophilic groove to the hydrophobic membrane core, in the channels the respective region remains closed, forming a protein-enclosed ion pore. It becomes more and more apparent that the scramblases of the family can perform both lipid scrambling and ion conduction, however, the relationship between the two processes is still unknown. It is disputed whether the observed ion conduction is a by-product of lipid scrambling, or whether the scramblases can adopt conformations similar to that observed for the channel homolog of the family, TMEM16A, and thereby mediate ion flux. Recent studies suggest that ion conduction and lipid scrambling might be separated by different conformational states. However, the proposed 'alternating pore-cavity' model required further experimental support. Study presented here addresses this question and how lipid movement is regulated using molecular dynamic simulations, cryo-electron microscopy and complementary scrambling assay.

Extending on their previous MD simulations, the authors were able to expand the data to better sample the different microstates present during simulation. Here, the authors were able to identify a partially closed-groove conformation, which involves a putatively lipid-induced repositioning of TM4, and argue that this conformation might be responsible for ion conduction. They suggest that hydrophobic interactions between TM3 and 4 play a key role in maintaining an open-groove scrambling-active conformation. Indeed, mutations of these residues result in severely compromised scrambling activity, while the channel activity is unaffected. The authors concluded that particularly L302A stabilizes the ion-conductive state of nhTMEM16. To further test their assumption, the authors proceeded with structure determination of nhTMEM16 L302A in lipid nanodiscs. The structure revealed nhTMEM16 in a state similar to those reported before for WT nhTMEM16 (Ca-bound intermediate in nanodiscs) and TMEM16F – with the lipid groove shielded from the membrane and potentially serving as an aqueous pathway for ions.

The study is a great example of the power of MD, as it allowed to identify key gating residues, postulate on mechanistic processes and because (some of) the conformations observed in trajectories match the experimentally structures reported recently and here. The conclusions are sound and strengthen the 'alternating pore-cavity' model proposed for TMEM16 scramblases. Overall, I want to congratulate the authors on the massive amount of data and the new exciting insights! I believe the study is of great interest to a broad community and meets the standards of Nature Communication. While I have some major comments that need to be addressed, I highly recommend the manuscript for publication.

Major comments:

1. While I very much welcome the hypothesis of a lipid-dependent gating, I have my doubts on how solid it is supported by the data. The assumption here is that the insertion of a lipid tail into the groove, triggers its closure leading to and stabilizing the occluded ion-conducting state. According to table 1 the authors observed the occluded state 7 times in their simulations. In Figure 2 they nicely plot the time-evolution of such a state and monitor the proximity of a lipid-tail, concluding that the structure shifts from microstate 1  4  3, while the lipid-tail was inserted in microstate 4. However, as the authors state themselves this lipid-tail insertion was a minor occurrence and happened predominately in microstate 4 and 7 which are for example not populated in the time-evolution shown for other subunits in suppl. Figure 4 (would be nice if we would see the data for all 7 events of occluded states). It is also very unclear to me how such a lipid could still be bound during the occluded state (according to suppl. Figure 4G and H and suppl. Figure 5 it seems like it). It seems to me that the occluded state can also be obtained without any

lipid-tail insertion. While I acknowledge that there is some correlation of lipid-tail insertion and the formation of the occluded state I do not see a clear pattern nor a requirement for it. Thus, I would at least tone down this statement.

2. The authors and other groups have shown that TMEM16 activity is greatly affected by the lipid composition. Yet this is entirely ignored throughout the manuscript although it is crucial at many stages. Then:

a) While MD simulations were performed at different lipid compositions (PE/PG or PC) the results were merged and are not discussed in this respect. Considering the proposed lipid-dependent gating mechanism it is tempting to assume that different lipid compositions might have a different impact on shifting the equilibrium between open and occluded states. If it's the insertion of the tail (and not headgroup) that stabilizes the occluded state the different lipids used might not have such a different impact, yet, could you comment what could be the difference between different lipid tails? Does it just happen more frequently in certain lipid compositions? Do you have any insight on that from MD?

b) the authors have recently shown (Lee, et al. *Biophysical Journal* 111, 1919–1924 (2016)) that nhTMEM16 is active in a lipid mixture of PE/PG/PC but loses its channel activity in a lipid mixture of PE/PG. Indeed, the scrambling and channel flux assays reported here were conducted in a PE:PG:PC mixture and show robust scrambling and ion conduction activity for the WT. Yet, the majority of the MD simulations and more importantly the nanodiscs lipid composition used for cryo-EM was PE/PG. I miss a clear discussion on this topic. In particular giving the main focus of the manuscript on the occluded potentially ion-conducting state that is claimed to be captured by MD and cryo-EM, I wonder how this fits. Since the protein has been biochemically shown to not be ion-conductive in 3:1 POPE/POPG, how do the author support to have found an ion-conducting state by MD and cryo-EM in this lipid composition?

3. If the mutation L302A favors the occluded conformation, why is the channel activity lower than for the WT? (See Figure 6). It is stated that the occluded state was a "rare conformation" in the WT, and that they could stabilize it with the mutant. But wouldn't you then expect a higher conduction for this mutant, since the equilibrium is shifted from to the open scrambling-conformation to the occluded and rare ion conductive state?

4. Since L302W was not suitable to trap the occluded potentially ion-conducting state how do you explain that L302W still shows ion conduction similar to L302A? (see previous paper Lee et al, 2018, *Nat Comm*).

5. Considering the number of nhTMEM16 structures determined so far and the significant differences observed dependent on the conditions (Ca-bound or Ca-free, detergent or nanodiscs, and which state was captured) it is crucial that the authors are more precise when refereeing and comparing states. I would thus highly advise the authors to revise this in the manuscript and always add this information when referring to one of the structures (conditions and state).

6. I am also surprised that the authors do not relate and discuss the MD findings and the occluded state in more depth with the different conformational states obtained in a previous publication on nhTMEM16. Especially because I believe they beautifully support each other. In fact, the conformational flexibility observed in MD goes in line with what was observed by cryo-EM of nhTMEM16 in nanodiscs in presence of calcium. Not just the three distinct states that were captured, but the fact that 50% of the particles show a weaker density for TM3 and TM4, highlighting their flexibility in solution. I also believe that the manuscript would benefit from a schematic mechanistic final image emphasizing their conclusions. In principle the authors agree that ion and lipids are conducted via different conformational states as proposed in the "alternating pore-cavity mechanism (ref 19).

7. I don't quite understand why an 'open' state of TMEM16A was modeled based on nhTMEM16. I do not see it adding anything valuable to the manuscript and would highly advise to take it out.

8. While it is not relevant for the regions interpreted in the manuscript it should be noted that it was recently found that 4WIS has multiple register shifts in the cytoplasmic domain. As 4WIS was used as reference and not the recently corrected structures (e.g. 6QM5), the model presented here still contains the same errors. Please correct them before depositions.

Minor comments:

(general remark upfront: it would have been highly appreciated if the authors would have included line numbers to guide the reviewer comments)

1. The fact that the subunits in the nhTMEM16 dimer adopt different conformations in the MD simulations is not discussed nor pointed out (very apparent in S7D). It has been shown, in particular for TMEM16A, that both subunits operate independently. Thus, I am not entirely surprised by it. However, how to explain that in cryo-EM the structures appears to be mostly symmetric?
2. Is it possible to provide some sort of numbers/percentage on the population of states? For example it is stated that microstate 3 showing the occluded state is a rare event. How many times from how many structures in total sampled did you observe it?
3. References appear to have been introduced in the manuscript without using an appropriate reference program. As a consequence, the wrong paper is sometimes cited (noticed often for papers ref 9-11 and 18-22) or there are even mistakes in the titles of manuscripts in their bibliography (e.g. ref22). Authors should carefully check citations for revision.
4. In the abstract it reads "A membrane-exposed hydrophilic groove in these proteins serves as the translocation pathway for both ions and lipids". As the authors conclude ions are most certainly conducted via a pore-like conformational state, thus not via a membrane-exposed hydrophilic groove as for lipids, the "membrane-exposed" is highly misleading.
5. Page 4 first sentence. Please note that the conformational change of TM6 in TMEM16A and TMEM16F go in opposite directions and not in both to the center of the groove.
6. A few sentences lower note that there is also a Ca-bound closed state of nhTMEM16, so not only of TMEM16K and TMEM16F.
7. Page 4 lower end. Misleading when stated about the occluded state: "This intermediate state in the scramblase, that encompasses a relatively small population of EM particles...". Please note that for TMEM16F this state was found technically in all particles. With regards to nhTMEM16 in nanodiscs Ca-bound about 50% of particles showed a high flexibility of TM3 and TM4, of the remaining 50%, 12% showed a distinct occluded state. How do these numbers relate to the populations seen in MD?
8. Page 16. "The core region is well resolved, with areas reaching a local resolution of 3Å..." I would recommend to tone this statement down by just stating "with regions at higher resolution. Then, if the final map only shows a global resolution of 4.0Å, the plotted local resolution seems a bit overestimated, which is a common known problem.
9. Page 16: "Notably, the density of the nanodisc membrane is strong at the lipid pathway, indicating that the membrane is not thinned and suggesting that the groove is in a non-scrambling conformation (Supplementary Figure 12I) 20." Without direct comparison to open structure of nhTMEM16 in similar conditions this statement is rather vague. I understand that such a structure is not available, and one can't simply compare nhTMEM16 in 2N2 vs nhTMEM16 in E3D1, as the latter nanodisc is smaller and this might result in a difference in bending. But to me this figure alone does not make it obvious that the membrane is not thinned in this case and that this is an indication that the conformation is not lipid-conductive.
10. Page 18: The RMSD between the cryo-EM L302A and the MD-microstate 3 is 1.9Å and not 1.2Å. In fact it is 1.2Å, and thus most similar to the nhTMEM16 Ca-bound nanodisc intermediate state (6QMA).
11. Page 20 and in the discussion it is stated that all Ca-bound TME16A structures are closed. While this is probably true it is misleading. The authors show in biochemically that L302A can conduct chloride ions, yet their structure does not show a wide enough pore to accommodate chloride ions (See figure 7G). The rationale is then that small side chains rearrangements are enough to widen the pore, which is logical and was the same argument postulated for TMEM16A.

12. Page 21/Discussion: Nice example of why it is important to specify the exact states of the referred nhTMEM16 structures. "In the structures of nhTMEM16 in absence of Ca, the groove is closed..." is technically wrong as the structure obtained in absence of Ca in detergent still shows an open groove (ref 19). This conformation would suit to discuss the proposed lipid-triggered conformational change to an occluded ion-conducting state. While I am not entirely convinced if the lipid-tail insertion is indeed required to obtain an occluded state (see above) it would perfectly fit with the authors logic: the occluded state would be less favored in a detergent environment, deprived of lipids.

13. I feel like the manuscript could be strengthened if the authors could make a final statement in the discussion/conclusion how the results presented here can be useful for other (particularly mammalian) scramblases, where the conformational spectrum is not yet sampled. Are similar residues present in other scramblases of the family? With this knowledge, will it be possible to predict how to convert any TMEM16 scramblase into a channel? Connecting this study to other TMEM16 scramblases involved in diseases will make it more interesting for a broader audience.

General remark on model representation of microstates. I would highly recommend to:

- a) only use one view and not rotate the protein when showing different microstates
- b) indicate membrane boundaries and label at least TM4 and 6
- c) indicate what the state represents (closed, fully open, etc). One suggestion would be to add three transparent boxes on top of the important interactions (E313/318/R432; T333-Y439 and V337-V447) and use a transparent color to indicate if they are 'closed' or 'open'. OR similar to S3 show a table overview with the distances and add threshold distance of what you call interacting/closed or separated/open.

Minor remarks/suggestions on Figures

Figure 1: Does this also include the WT(ensemble)?

Figure 3: Since this is still all WT, why suddenly name the microstates a,b,c? The figure legend reveals they actually represent microstate 1, 4 and 3. Why not use that from the start?

Figure 5: Similar to comment on Figure 3, why not stick to the same nomenclature of microstates as introduced for the WT. To me it doesn't seem that different conformations were obtained with the mut, but just the abundance of populations were shifted. If I understood it correctly a would be state 3, b=1 and c=7? You could use a subscript to indicate they derive from the mutant MD simulations. Yet it would help the reader to stick to one nomenclature of conformational states. For panel C include a line for the threshold of when something is considered open or closed (around 8Å as suggested by the authors). To guide the reader it might also be good to use a different color for the models to distinguish it from the WT in the previous figures?

Figure 7: Panel D: perhaps add a dashed line to indicate the suggested upward movement of TM3.

Legend of panel G: To avoid confusion stick to pore radius or diameter. Why not show an actual superposition of WT and mutant of the groove like in panel C.

Figure 8B: Here (or at least as supplementary) would be nice to see a superposition of the cryo-EM to the nanodisc Ca-bound intermediate nhTMEM16 structure (6QMA).

Figure 9A: for a proper comparison it would be better to show the exact same (superimposed) view of the different models

S1: see major comment on figures and how to best represent the microstates. Panel C: good example on what is actually 100% in the plotted frequency? About how many structures with broken EC of how many structures in total are we talking about?

S2: Don't show the lipid in the same color as the residues. If these plots show the 7 events of flipping and it microstate 5 has been introduced as the only fully-open state that can actually flip lipids, how does it come that microstate 5 seems only to be reached in 4 out of the 7 trajectories? Indicate in figure legend as for S4 that the trajectories are plotted on the 2D landscape shown in Fig.1.

S3, why not write the actual residues on top of each column? Include threshold on when something is considering interacting/closed or broken. Why are the T33-Y439 and E313/318-432

distances not included?

S4: Why not show it for all 7 events where an occluded state was observed? For WT6 and WT4 (panels b,c,e,f) the V337-447 distances do not reach below 8Å which was the indicated hallmark for microstate 3, are they really microstate 3 occluded then? Also see comment in major remarks about observed results on proximity of lipids and reaching microstate 4. Does not look consistent and would be interesting to see how the other 3 trajectories of missing occluded states would look like.

S5: Probably my problem but I do not understand what fraction of lipid penetration means? How deep it is in the cavity?

S6: include a line indicating the >3.5Å which the authors defined as broken interaction. Panel C: Why not include here also directly the same plot for L302W and L302A? The characteristic broken TM3-4 interaction claimed to include the occluded state is stronger represented in microstate 7 and to some extent 6. Any comments on this?

S8: Panel A: please include also a 90degree rotation as shown for the WT to see how TM4 bents.

Panel E: I understand from the graph that actually 70% of L302W do not have any lipid tails inserted, which is the opposite of what is stated in the manuscript. I am reading the graph wrong?

S11: The gel with L347 is missing, consider including for consistency

S12: Panel H. Perhaps also show a zoom-in by 9- degree rotated superposition of the grove in both structures.

S14: meshes are barely visible.

S15+16: be more precise in figure legend describing which nhTMEM16 structure was used (Ca-bound in nanodiscs, intermediate, etc...)

Once again, congratulation on the new study and I hope my comments help to further strengthen the manuscript.

Reviewer #2:

Remarks to the Author:

This paper from the Weinstein and Accardi labs seeks to understand the structural features that determine the ion and lipid conducting properties of the TMEM16 family of proteins. While there is growing interest in this family of proteins because of their disease relevance, the basic biophysics of these proteins remains enigmatic partly because of their dual function as lipid scramblases and/or ion channels. Making very impressive use of molecular dynamics simulations, biochemical analysis of lipid scrambling and ion flux, and cryo-EM these investigators provide fundamental insights into this problem. The work is truly a tour-de-force. The paper is dense with data that is interpreted fairly to yield a solid conclusion. The paper is remarkably clearly presented, especially considering the density and complexity of the data. I am not qualified to evaluate the details of the tICA approach, but the overall design seemed reasonable.

1. The authors could strive to highlight the broader significance of their work. As it stands, the details are the strength of the paper, but they might not be so interesting to those outside the small group of biophysicists who work on these TMEM16 proteins. The broader implications are relatively obvious and could be developed without much difficulty.

2. I was not convinced by the idea that lipid tail insertion was a trigger for the conformational change to the ion-conducting state. It seems that this could be presented more clearly and the delineation between correlation and causality addressed more rigorously. In Fig. 2B,C, the correspondence between lipid insertion and conformational change is poor in the time domain. It might help to know more atomistic details about what exactly is happening during lipid insertion. If the transition to microstate 3 occurs after the lipid has departed, what evidence is there that the lipid has any role in this process? Supplementary Fig. 5 seems a crucial issue, but I felt that these

data could have been crunched in a more convincing way. It would also help to see representative images of the lipid tails and water in the conduction pathway during the trajectory to microstate 3.

3. The authors sweep under the rug the fact that the ion flux assay measures Cl conductance but yet the molecular models do not have a pore dilated enough to conduct Cl. This discrepancy is brushed aside by saying that there is probably another state with a slightly dilated pore. While I agree that this is a logical presumption, it deserves more than a sentence in passing. Is there any evidence for the existence of such a state in the long MD simulations?

Reviewer #3:

Remarks to the Author:

The authors explore the relationship between ion conduction, lipid flipping, and protein conformational state in the fungal nhTMEM16 scramblase. Based on simulations, the authors determined that TM4 can undergo a transition from a fully open structure to a partially closed structure that stops lipid flipping. They outline the importance of lipids in this event as well as the importance of specific residues, in particular L302, which helps stabilize a TM3-TM4 interaction that becomes disrupted to close the lipid groove. Based on the importance of L302 for keeping the groove open, they made the mutation to a small hydrophobic residue (L302A) and then solved the cryo-EM structure in nano-disc with Ca²⁺ bound. They observe a closed groove that still forms a pore through the membrane large enough for small ions but not exposed to lipid, as suggested by the simulations. Overall this work is impressive with a new structure, extensive simulations, and supporting functional data. I am supportive of this work, and I think that it is a step forward to understanding how scramblases also conduct ions and the conformational changes that allow one or the other to occur. That said, you will see that I have many questions, and I think that the presentation needs to be clarified in some places.

Major concerns

My two biggest concerns are the following:

1. The authors have not convinced me that the lipid tail insertion is the key to groove closure. I am left with the feeling that this is an anecdotal event that may or may not be the primary pathway for groove closure (and opening??). I would like to be convinced.
2. Are the TM3-TM4 motions revealed by the simulations and the L302A structure on path for wild type nhTMEM16 function, or are they specific to the L302A mutant? To this end, I would like the authors to compare TM3-4 in their new structure to the new Kaleinkova Elife structures. Are they seeing the same kind of relative motions in their structures, or is TM3-4 moving differently.

Table 1. While I realize that these results are from a prior publication, it is interesting that the authors observe 7 in to out flips in the ACEMD set of simulations (called WTensemble here), and no inward flips. Do they have a feeling for why this happened? Also, would the authors comment on whether they think the ACEMD dynamics engine (barostat, thermostat, other implementations) somehow is biased towards seeing flipping, while Anton2 might suppress this? Is it all related to the propensity for the groove to fully open or the lipid systems? Have the authors used NAMD, Amber or Gromacs with the CHARMM36 force field? I realize that equilibration for Anton2 was with NAMD, but these were short simulations.

The authors use distances between residues in the extracellular site and residues near the mid-plane of the groove (distance between TM4 and TM6) to judge the openness of the groove. Figure 2A shows the first two tICAs – how do these project onto your original set of CVs? How correlated are your CVs? I expect that they are highly correlated. For instance, is tICA one the extracellular site opening and tICA 2 the mid-plane opening?

I very much like the kind of analysis that the authors present in Figure 2; however, I have a hard

time teasing apart the lipid degrees of freedom from the protein degrees of freedom. For instance, the authors discuss a lipid flipping event and the corresponding microstates in Fig 2A, but I got lost very quickly about whether this was a discussion about 1 lipid flipping event or a description of many events that all behave similarly. How biased are the protein CVs by where permeating lipids are? To this end, are the CVs computed with c-alpha-c-alpha distances or side chain distances (I am pretty sure they are all c-alpha-c-alpha)? Nonetheless, side chain distances would constitute a large motion in tICA space during a lipid permeation event that separates residues (say at the extracellular site), while having very little influence on the slower degrees of freedom of the protein (i.e. helix-helix separation).

The authors explain at the top of page 9 how extracellular residues E313 and R432 play an important role in lipid flipping and release, and they cite a past paper where they discuss these residues and another one. They should also cite the Bethel and Grabe PNAS paper, where these two first residues were first highlighted.

From a structural point of view, the movements of TM3 and TM4 in Figure 1A in microstates 1 versus 3 are very appealing. This is how I imagine lipid gating likely happens. In the Markov State Modeling literature, is it better to call these states macrostates rather than microstates? I do not remember the criteria used by the Pande lab (and others) to distinguish these two ideas, but it is worth looking into.

Several places through the manuscript the authors use the word enhanced. I do not see how these are enhanced simulations in the classic/standard view of enhanced sampling. I would remove this term, or if I missed it, please correct me.

In Figure 2A, please provide the axes values so that the reader can appreciate where these points lie in the phase plane of Figure 1A. This figure is very interesting, and I find this kind of molecular level detail explaining a configurational transition enlightening; however, this is all based on a single trajectory. There is nothing to suggest that this is the primary pathway to groove closure. It is said that microstates 4 and 7 experience tail entry in several cases, but how many other examples like WT6 are there? Can you provide statistics about pore closure? If not, the authors should present some kind of a PMF or energy calculation to support the generality of their claim in Figure 2.

Figure 3 is not really described in the main text. Is panel C defined by the box in panel A? Likewise, are panels B and D above and below this boxed region?

The simulations in Figure 4 suggest that partial closure of the gate is accompanied by separation of L302 on TM3 from 343/347 in TM4. The authors should look at the homologous residues across all known structures (scramblers and not as well as their new structure reported here) and show that partial or full closure of the groove does involve greater distances between these residues. This is important because it gets at whether TM3-4 do or do not move as a unit during gating. I now realize that this is done later in manuscript with regard to the new mutant structure – very nice, but I would like to see it compared to all structures.

The experiments in Figure 5 are impressive. Were any mutants at V447 made (here or in other studies), and if so, what do they do? Is the scramblase activity as critical at this site?

On page 16, the authors state:

“The nhTMEM16 L302A/nanodisc complex is bent along the dimer cavity (Supplementary Figure 12I), consistent with recent findings on how the TMEM16 scramblases affect the surrounding membrane 19,20.”

I appreciate that the authors are referring to recent structural work in nano-disc that has experimentally revealed the shape of the surrounding membrane, but it might be appropriate to cite Bethel and Grabe as they were the first to predict this several years prior with computation.

The structural work in Figure 7 is very nice, and the figure illustrates all of the concepts discussed throughout the manuscript nicely. I appreciate the overlay of the recent structures from the Kaleinkova nhTMEM16 manuscript. One thing I noticed is that while TM6 is in a similar place in all 4 structures, the position of TM3 is similar for all except the most recent mutant. Does this lead us to think that the mutant is adopting a conformation that is not frequently sampled in the WT protein?

The pore electrostatics profile work is nice. I especially like the many curves superposed on each other taken from the simulations (Fig. 9F).

The work reported here may be used to support a model that ions and lipids permeate via different states. The authors don't make any comment about lipid-ion co-permeation as shown in the Elife paper by Tajkhorshid and Hartzell. I think that the authors should at least comment on this even if they can't make any solid claims one way or the other. In particular the 3 models put forth at the end of the Kaleinkova Elife paper are useful for framing this.

Minor

At the top of page 4, first line. "it occupies a more central" – what is "it"? Are you referring to Ca²⁺ here?

First paragraph of figure 4:

"...giving rise to the Ca²⁺-bound closed conformation seen in the TMEM16K and TMEM16F scramblases 21"

Reference 21 is for K only, not F. Please insert F citation.

On page 4, the authors state:

"In the TMEM16A chloride channel, the groove serves as an ion-selective pore ..."

I would not call this a groove, and I think that many others would not either. I think some care should be taken at a few points in the manuscript to not call pores grooves.

Again in the middle of page 4,:

"...wider groove compared to that in the Ca²⁺-bound TMEM16A channel"

I think that care should be given to calling structures grooves versus pores. Imagine that a conformational change opens a pore to form a groove. I still think that one should not call the pore a groove. I also realize that the authors are careful about this at other points in the manuscript.

At the bottom of Page 16 the authors use the phrase "extracellular vestibule" without defining it.

Throughout the manuscript "microstate" and "alanine" are often capitalized when they should not be. For instance on page 24, "L302 to Alanine" should be "L302 to alanine".

Reviewer #1 (Remarks to the Author):

TMEM16 family of membrane proteins displays a remarkable functional dichotomy, with some members working as selective anion channels, and others primarily as lipid scramblases with reported non-selective channel activity. Structural work conducted on TMEM16 channels and scramblases allowed to identify architectural differences between the two functional branches of the family – while the scramblases upon activation expose a hydrophilic groove to the hydrophobic membrane core, in the channels the respective region remains closed, forming a protein-enclosed ion pore. It becomes more and more apparent that the scramblases of the family can perform both lipid scrambling and ion conduction, however, the relationship between the two processes is still unknown. It is disputed whether the observed ion conduction is a by-product of lipid scrambling, or whether the scramblases can adopt conformations similar to that observed for the channel homolog of the family, TMEM16A, and thereby mediate ion flux. Recent studies suggest that ion conduction and lipid scrambling might be separated by different conformational states. However, the proposed ‘alternating pore-cavity’ model required further experimental support. Study presented here addresses this question and how lipid movement is regulated using molecular dynamic simulations, cryo-electron microscopy and complementary scrambling assay.

Extending on their previous MD simulations, the authors were able to expand the data to better sample the different microstates present during simulation. Here, the authors were able to identify a partially closed-groove conformation, which involves a putatively lipid-induced repositioning of TM4, and argue that this conformation might be responsible for ion conduction. They suggest that hydrophobic interactions between TM3 and 4 play a key role in maintaining an open-groove scrambling-active conformation. Indeed, mutations of these residues result in severely compromised scrambling activity, while the channel activity is unaffected. The authors concluded that particularly L302A stabilizes the ion-conductive state of nhTMEM16. To further test their assumption, the authors proceeded with structure determination of nhTMEM16 L302A in lipid nanodiscs. The structure revealed nhTMEM16 in a state similar to those reported before for WT nhTMEM16 (Ca-bound intermediate in nanodiscs) and TMEM16F – with the lipid groove shielded from the membrane and potentially serving as an aqueous pathway for ions.

The study is a great example of the power of MD, as it allowed to identify key gating residues, postulate on mechanistic processes and because (some of) the conformations observed in trajectories match the experimentally structures reported recently and here. The conclusions are sound and strengthen the ‘alternating pore-cavity’ model proposed for TMEM16 scramblases. Overall, I want to congratulate the authors on the massive amount of data and the new exciting insights! I believe the study is of great interest to a broad community and meets the standards of Nature Communication. While I have some major comments that need to be addressed, I highly recommend the manuscript for publication.

We thank Dr. Paulino for this enthusiastic evaluation of our manuscript, and the thoughtful comments that we address in detail below.

Major comments:

1. While I very much welcome the hypothesis of a lipid-dependent gating, I have my doubts on how solid it is supported by the data. The assumption here is that the insertion of a lipid tail into the groove, triggers its closure leading to and stabilizing the occluded ion-conducting state. According to table 1 the

authors observed the occluded state 7 times in their simulations. In Figure 2 they nicely plot the time-evolution of such a state and monitor the proximity of a lipid-tail, concluding that the structure shifts from microstate 1  4  3, while the lipid-tail was inserted in microstate 4. However, as the authors state themselves this lipid-tail insertion was a minor occurrence and happened predominately in microstate 4 and 7 which are for example not populated in the time-evolution shown for other subunits in suppl. Figure 4 (would be nice if we would see the data for all 7 events of occluded states). It is also very unclear to me how such a lipid could still be bound during the occluded state (according to suppl. Figure 4G and H and suppl. Figure 5 it seems like it). It seems to me that the occluded state can also be obtained without any lipid-tail insertion. While I acknowledge that there is some correlation of lipid-tail insertion and the formation of the occluded state I do not see a clear pattern nor a requirement for it. Thus, I would at least tone down this statement.

We were led to our conclusion that groove occlusion is related to lipid tail insertion by the data from the MD simulations showing that in every instance of the observed groove closure the lipid tail had penetrated into the groove, i.e., groove closure is not observed unless a lipid tail had been inserted there first. As described in the manuscript, once the resulting occluded conformation of the groove is established it can remain stable also in the absence of the contact with the lipid tail.

To illustrate this point better, we follow the suggestion and now present the data for all 7 instances of groove closure we observed for the wild type system (see revised main text pages 11-12, and revised Suppl. Figure 4). Thus, in the revised Suppl. Figure 4 we plot in panels A-R the results from 6 trajectories; the 7th is in Fig 2 as in the original manuscript. Note that in some cases the groove closure is transient (panels J-R), and in some simulations the occlusion takes place during the final stages of the trajectories (panels B, E, H, and C, F, I). To facilitate reading of the time-evolution trends in Suppl. Figure 4 and Fig. 2, we demarcate on the V337-V447 distance plots the distance of 8.5Å that we use as a threshold to classify a conformation as occluded (horizontal dashed line). This threshold is based on the time-evolution of the V337-V447 distance in our longest, 15 μ s MD simulation (Figure 2B) in which the V337-V447 distance fluctuates around \sim 8.5Å value once the occlusion occurs.

As can be seen in Suppl. Figure 4, all the trajectories progress dynamically towards Microstate 3. Furthermore, in all cases, even when the occlusion is transient, the hydrophobic tails of a single lipid molecule coordinate V337 and V447 residues. As we explain in the manuscript, physics considerations support the inference that the presence of the lipid tails can facilitate attractive interactions between the two hydrophobics, V337 and V447, thereby promoting the groove closure.

We note that it is *not* the case that a groove closure occurs every time a lipid tail inserts. But, as stated, we have not observed an event of groove closure without the lipid tail being inserted in the groove. Nevertheless, we agree with the Reviewer that even taken together, these observations do not prove our stated conclusion definitively, as they do not completely rule out the possibility that the occlusion takes place without lipid participation and that the lipid tail insertion is not required *per se* but could serve as a facilitator of the closure. Therefore, we have revised the manuscript accordingly throughout to address this point (e.g. see Discussion, page 23).

As to the question of the mode of lipid partitioning in the groove while the groove is occluded, indeed in our simulations we observed different ways that lipid tails can occupy the groove,

varying in the extent of the penetration (i.e., what portion of the lipid tail is inserted) and the level of insertion (i.e., which part of the groove is occupied by the lipid tail). Some of these variations are illustrated in the revised Supp. Figure 4 (panels S and T) and are reproduced in Figure 1 below. As can be seen, one tail of the lipid stays just below the V337-V447 pair and points into the membrane, while the second tail is inserted into the groove. This second tail can insert fully into the groove (as in Fig. 1A below), or it can present only its top portion (connecting to the lipid backbone) while the lower part of the tail goes above the V337-V447 pair pointing into the membrane.

Figure 1: Various modes of lipid tail insertion into the groove leading to the groove closure.

To better illustrate how lipid-protein interaction can facilitate the groove closure we have also included a Supplementary Movie showing the occlusion process. We hope that with these additions and clarifications we manage to address the important points raised by the reviewer in this comment.

2. *The authors and other groups have shown that TMEM16 activity is greatly affected by the lipid composition. Yet this is entirely ignored throughout the manuscript although it is crucial at many stages. Then:*

a) While MD simulations were performed at different lipid compositions (PE/PG or PC) the results were merged and are not discussed in this respect. Considering the proposed lipid-dependent gating mechanism it is tempting to assume that different lipid compositions might have a different impact on shifting the equilibrium between open and occluded states. If it's the insertion of the tail (and not headgroup) that stabilizes the occluded state the different lipids used might not have such a different impact, yet, could you comment what could be the difference between different lipid tails? Does it just happen more frequently in certain lipid compositions? Do you have any insight on that from MD?

The groove closure events occur in both lipid compositions during our MD simulations. However, the statistics are not sufficient to distinguish whether the frequency of occurrence of the transition or the stability of the conformation depend on the lipid composition. Further work will be needed to evaluate this point.

b) the authors have recently shown (Lee, et al. Biophysical Journal 111, 1919–1924 (2016)) that nhTMEM16 is active in a lipid mixture of PE/PG/PC but loses its channel activity in a lipid mixture of PE/PG. Indeed, the scrambling and channel flux assays reported here were conducted in a PE:PG:PC mixture and show robust scrambling and ion conduction activity for the WT. Yet, the majority of the MD simulations and more importantly the nanodiscs lipid composition used for cryo-EM was PE/PG. I miss a clear discussion on this topic. In particular giving the main focus of the manuscript on the occluded potentially ion-conducting state that is claimed to be captured by MD and cryo-EM, I wonder how this fits. Since the

protein has been biochemically shown to not be ion-conductive in 3:1 POPE/POPG, how do the author support to have found an ion-conducting state by MD and cryo-EM in this lipid composition?

We thank the reviewer for raising this critical point. As we noted in our manuscript, the minimal internal radius of the ion pore defined by the L302A mutant is sufficient to allow passage of Na⁺ and K⁺, but it is too narrow for Cl⁻ permeation. Further, our MD simulations of the L302A cryoEM structure suggest that the pore can become sufficiently wide to allow Cl⁻ movement due to subtle rearrangements of the pore-lining side chains. It is possible that specific interactions of lipid headgroups with residues in the TM4 and TM6 helices, and/or changes in overall membrane properties, might favor these small rearrangements, so that the pore becomes more or less conductive depending on the specific lipid composition. Indeed, in our previous manuscript (Lee et al., Nat Comms, 2018) we reported that interactions between lipid headgroups and residues lining the lipid groove are important to enable the adoption of a lipid-conductive conformation of the pathway. Importantly, in our MD simulations we observe groove closure events in both POPE/POPG mixtures and in POPC membranes. The sampling of these events in the trajectories is not sufficiently large to permit a statistically significant of the likelihood for closure in these different lipid compositions. Finally preliminary experiments in the lab suggest that the lack of channel activity in POPE/POPG liposomes might reflect -at least in part- the presence of different lipid phases in this mixture (Falzone, Lee and Accardi, unpublished), as its melting temperature (T_m) is close to room temperature. It was reported recently that reconstitution in nanodiscs lowers the melting temperature of lipids (Martinez et al., Chemphyschem, 2017), raising the possibility that the structure of L302A nhTMEM16 in POPE:POPG nanodiscs might indeed reflect a conductive conformation. Further experiments are needed to clarify this possibility. We now discuss this important point on page 25.

3. If the mutation L302A favors the occluded conformation, why is the channel activity lower than for the WT? (See Figure 6). It is stated that the occluded state was a "rare conformation" in the WT, and that they could stabilize it with the mutant. But wouldn't you then expect a higher conduction for this mutant, since the equilibrium is shifted from to the open scrambling-conformation to the occluded and rare ion conductive state?

Our data do not rule out that in nhTMEM16 ion translocation events can occur also through the scrambling-competent conformation. What our results show is that the occluded state we describe can still conduct ions but not lipids and that the lipid scrambling activity is significantly affected by this mutation which has a structural propensity to adopt an occluded groove.

4. Since L302W was not suitable to trap the occluded potentially ion-conducting state how do you explain that L302W still shows ion conduction similar to L302A? (see previous paper Lee et al, 2018, Nat Comm).

We do not know how the conformational changes seen in the L302W mutant simulations relate to the functional mechanism. It will be intriguing to explore this aspect in future studies.

5. Considering the number of nhTMEM16 structures determined so far and the significant differences observed dependent on the conditions (Ca-bound or Ca-free, detergent or nanodiscs, and which state was

captured) it is crucial that the authors are more precise when refereeing and comparing states. I would thus highly advise the authors to revise this in the manuscript and always add this information when referring to one of the structures (conditions and state).

We have carefully reviewed the text to address this issue.

6. I am also surprised that the authors do not relate and discuss the MD findings and the occluded state in more depth with the different conformational states obtained in a previous publication on nhTMEM16. Especially because I believe they beautifully support each other. In fact, the conformational flexibility observed in MD goes in line with what was observed by cryo-EM of nhTMEM16 in nanodiscs in presence of calcium. Not just the three distinct states that were captured, but the fact that 50% of the particles show a weaker density for TM3 and TM4, highlighting their flexibility in solution. I also believe that the manuscript would benefit from a schematic mechanistic final image emphasizing their conclusions. In principle the authors agree that ion and lipids are conducted via different conformational states as proposed in the “alternating pore-cavity mechanism (ref 19).

We thank the reviewer for this important observation. We have revised and edited the manuscript to make better connections between our findings and those from cryo-EM. Our findings identify a new functional state of nhTMEM16 that can support the transfer of ions but not of lipids.

We also considered the suggestion of presenting a schematic summarizing mechanistic findings from our work and have added Figure 10 in the revised manuscript (see also revised text on page 26). This Figure shows the known spectrum of structures sampled by Ca²⁺-bound nhTMEM16 that includes states labeled as *closed*, *intermediate*, and *open* states (PDBID: 6QMB, 6QMA, 6QM9, respectively), as well as the ion-conductive state (L302A structure, PDBID: 6OY3), and lipid-conductive state (from MD simulations in our previous paper, Lee et al, Nat Comms, 2018).

7. I don't quite understand why an 'open' state of TMEM16A was modeled based on nhTMEM16. I do not see it adding anything valuable to the manuscript and would highly advise to take it out.

Following the Reviewer's suggestion, we have removed the results pertaining to an 'open' state TMEM16A.

8. While it is not relevant for the regions interpreted in the manuscript it should be noted that it was recently found that 4WIS has multiple register shifts in the cytoplasmic domain. As 4WIS was used as reference and not the recently corrected structures (e.g. 6QM5), the model presented here still contains the same errors. Please correct them before depositions.

We thank the reviewer for pointing this out. We have rebuilt the cytosolic domain of the L302A mutant of nhTMEM16 using the new 6QM5 structure as a template.

Minor comments:

(general remark upfront: it would have been highly appreciated if the authors would have included line numbers to guide the reviewer comments)

9. *The fact that the subunits in the nhTMEM16 dimer adopt different conformations in the MD simulations is not discussed nor pointed out (very apparent in S7D). It has been shown, in particular for TMEM16A, that both subunits operate independently. Thus, I am not entirely surprised by it. However, how to explain that in cryo-EM the structures appears to be mostly symmetric?*

We thank the reviewer for raising this point. Our initial cryoEM dataset for the L302A mutant of nhTMEM16 is comprised of ~XXX particles, but the final reconstruction only contains XXX particles. It is possible that the asymmetric dimers are not sufficiently common to be separated in distinct classes during cryoEM processing. Furthermore, if we assume for simplicity that the dimers are independent and that the groove can adopt only 5 distinct Ca²⁺-bound conformations (closed; intermediate; ion conductive; membrane open; lipid conductive), then we would expect up to 2⁵=32 possible types of dimers. The fact that experimentally we observe only one major class, suggests that this is the most probable conformation of the groove; other classes might be present but at an abundance that is not sufficiently high to be resolved.

10. *Is it possible to provide some sort of numbers/percentage on the population of states? For example it is stated that microstate 3 showing the occluded state is a rare event. How many times from how many structures in total sampled did you observe it?*

On theory grounds, the most meaningful way to quantify the occurrence of the occluded state in our MD simulations, is to report the number of trajectories in which this state was observed in the total number of simulations. These are the numbers given in the original manuscript. Statistical analyses can be applied, of course, but such data could be misleading because, as shown in and Fig. 2 and Supp. Fig. 4, in some trajectories this state is long-lived, in some it is transient or emerges towards the end of the simulation. Thus, although the analysis is based on long MD trajectories, the populations of different states may not have converged to complete equilibrium values. Therefore, a probability distribution calculation would not be accurate. What remains clear, however, is the much-increased occurrence of the occluded-groove state in the mutant compared to the wild type scramblase.

11. *References appear to have been introduced in the manuscript without using an appropriate reference program. As a consequence, the wrong paper is sometimes cited (noticed often for papers ref 9-11 and 18-22) or there are even mistakes in the titles of manuscripts in their bibliography (e.g. ref22). Authors should carefully check citations for revision.*

Thank you for point this out. We have carefully reviewed the references and we believe all the proper citations now should be in place.

12. *In the abstract it reads "A membrane-exposed hydrophilic groove in these proteins serves as the translocation pathway for both ions and lipids". As the authors conclude ions are most certainly conducted via a pore-like conformational state, thus not via a membrane-exposed hydrophilic groove as for lipids, the "membrane-exposed" is highly misleading.*

This sentence in the Abstract introduces the groove and its functional properties as it is well-accepted in the community. We describe the circumstances of the occlusion of the groove and the conclusion that this leads to a conduit for ions via a pore-like conformational state.

13. Page 4 first sentence. Please note that the conformational change of TM6 in TMEM16A and TMEM16F go in opposite directions and not in both to the center of the groove. A few sentences lower note that there is also a Ca-bound closed state of nhTMEM16, so not only of TMEM16K and TMEM16F.

Thank you for pointing out this omission. This has been corrected in the revised manuscript (pages 3-4).

14. Page 4 lower end. Misleading when stated about the occluded state: "This intermediate state in the scramblase, that encompasses a relatively small population of EM particles...". Please note that for TMEM16F this state was found technically in all particles. With regards to nhTMEM16 in nanodiscs Ca-bound about 50% of particles showed a high flexibility of TM3 and TM4, of the remaining 50%, 12% showed a distinct occluded state. How do these numbers relate to the populations seen in MD?

We are grateful for this important comment. In the revised manuscript (page 4) we present this description separately for nhTMEM16 and mTMEM16F citing the appropriate papers. As to the question about the populations observed in MD, please see our response to Question 10 above.

15. Page 16. "The core region is well resolved, with areas reaching a local resolution of 3Å..." I would recommend to tone this statement down by just stating "with regions at higher resolution. Then, if the final map only shows a global resolution of 4.0Å, the plotted local resolution seems a bit overestimated, which is a common known problem.

As suggested, we changed the text to: "The core transmembrane region is well resolved, with areas of higher local resolution..."

16. Page 16: "Notably, the density of the nanodisc membrane is strong at the lipid pathway, indicating that the membrane is not thinned and suggesting that the groove is in a non-scrambling conformation (Supplementary Figure 12I) 20." Without direct comparison to open structure of nhTMEM16 in similar conditions this statement is rather vague. I understand that such a structure is not available, and one can't simply compare nhTMEM16 in 2N2 vs nhTMEM16 in E3D1, as the latter nanodisc is smaller and this might result in a difference in bending. But to me this figure alone does not make it obvious that the membrane is not thinned in this case and that this is an indication that the conformation is not lipid-conductive.

We agree, and have removed the sentence.

17. Page 18: The RMSD between the cryo-EM L302A and the MD-microstate 3 is 1.9Å and not 1.2Å. In fact, it is 1.2Å, and thus most similar to the nhTMEM16 Ca-bound nanodisc intermediate state (6QMA).

We are sorry for the confusion. Thus, in the manuscript we state "Overall, the cryo-EM structure of the L302A is in remarkable agreement with the computationally predicted model of this mutant, as underscored by the low RMSD (~1.2 Å) over the backbone atoms of the transmembrane helices for one subunit between the two structures". Perhaps, the confusion is due to the fact that in Table S1 we report RMSD of 1.9Å between L302A and WT_inter (which is the intermediate state assumed by the wild type protein). But on Page 18 of the original manuscript, we are comparing L302A

structures determined from cryo-EM and predicted by MD. The RMSD between these structures is indeed 1.2Å. We hope this is now clear.

18. Page 20 and in the discussion it is stated that all Ca-bound TME16A structures are closed. While this is probably true it is misleading. The authors show in biochemically that L302A can conduct chloride ions, yet their structure does not show a wide enough pore to accommodate chloride ions (See figure 7G). The rationale is then that small side chains rearrangements are enough to widen the pore, which is logical and was the same argument postulated for TMEM16A.

Thank you for pointing out that this could be confusing. We agree and have revised the manuscript (see page 22) to address this comment.

19. Page 21/Discussion: Nice example of why it is important to specify the exact states of the referred nhTMEM16 structures. "In the structures of nhTMEM16 in absence of Ca, the groove is closed..." is technically wrong as the structure obtained in absence of Ca in detergent still shows an open groove (ref 19). This conformation would suit to discuss the proposed lipid-triggered conformational change to an occluded ion-conducting state. While I am not entirely convinced if the lipid-tail insertion is indeed required to obtain an occluded state (see above) it would perfectly fit with the authors logic: the occluded state would be less favored in a detergent environment, deprived of lipids.

Thank you for the comment. We now mention (pages 3 and 22) that the Ca²⁺-free structure was obtained in detergent. As to the effect of inserted detergent in the EC side, it is actually an interesting point and indeed fits with our observation that when lipid tails occupy the EC side of the groove (as in Microstate 7; see Figure 2 below and our response to Question 25 below), the groove remains open.

20. I feel like the manuscript could be strengthened if the authors could make a final statement in the discussion/conclusion how the results presented here can be useful for other (particularly mammalian) scramblases, where the conformational spectrum is not yet sampled. Are similar residues present in other scramblases of the family? With this knowledge, will it be possible to predict how to convert any TMEM16 scramblase into a channel? Connecting this study to other TMEM16 scramblases involved in diseases will make it more interesting for a broader audience.

We agree fully with the importance of this consideration and are devoting special attention to this direction in our current research, investigating common and divergent mechanisms in fungal and mammalian TMEM16s. We have added a concluding statement in Discussion along these lines (pages 26-27).

21. General remark on model representation of microstates. I would highly recommend to:

a) only use one view and not rotate the protein when showing different microstates

b) indicate membrane boundaries and label at least TM4 and 6

c) indicate what the state represents (closed, fully open, etc). One suggestion would be to add three transparent boxes on top of the important interactions (E313/318/R432; T333-Y439 and V337-V447) and

use a transparent color to indicate if they are 'closed' or 'open. OR similar to S3 show a table overview with the distances and add threshold distance of what you call interacting/closed or separated/open.

We are grateful for these thoughtful suggestions. Following (b), we have added labels for TMs 4 and 6 in all the microstate representations. Unfortunately, showing membrane boundaries would complicate matters significantly because the membrane undergoes strong deformations around the groove area, and these are locally different due to conformational differences, so showing these in any useful detail would reduce much the clarity of the images of the microstate models. Following (c), we have added in Figure 1 histograms of V337-V447 distance for microstates 1 and 3 (similar to what we have done for the L302A system in Figure 5). The rest of the histograms are shown in Supp. Figure 3 as before. We had described the EC gating mechanism in detail in our previous paper (Lee et al, Nat Comm), we put the recurring details in the Supporting figures so as not to burden the main text of the current manuscript. We believe that all the quantitative information (including threshold distance for the EC gates) given in the labels of these figures are sufficient to provide a complete description of the structural characteristics of the various regions on the tICA space. As we describe in response to Question 1 above we have added to Figure 1 (and all the other figures that report on V337-V447 distance) the threshold lines for the V337-V447 distances indicating the cutoff we consider for the occluded conformation.

Minor remarks/suggestions on Figures

22. *Figure 1: Does this also include the WT(ensemble)?*

As stated in the caption of Figure 1 ("*2--D landscape representing all the trajectories of the WT nhTMEM16 protein (Table 1)*") and also in the text, the analysis does indeed include the WT(ensemble) simulations.

23. *Figure 3: Since this is still all WT, why suddenly name the microstates a,b,c? The figure legend reveals they actually represent microstate 1, 4 and 3. Why not use that from the start?*

Thank you – this was done.

24. *Figure 5: Similar to comment on Figure 3, why not stick to the same nomenclature of microstates as introduced for the WT. To me it doesn't seem that different conformations were obtained with the mut, but just the abundance of populations were shifted. If I understood it correctly a would be state 3, b=1 and c=7? You could use a subscript to indicate they derive from the mutant MD simulations. Yet it would help the reader to stick to one nomenclature of conformational states. For panel C include a line for the threshold of when something is considered open or closed (around 8Å as suggested by the authors). To guide the reader it might also be good to use a different color for the models to distinguish it from the WT in the previous figures?*

We have added a line representing 8.5Å threshold we use for the V337-V447 distance (text has been modified accordingly). We chose not to change the nomenclature of microstates since we think this may confuse a reader because Figure 5 relates to Supp. Figure 9 in which the same "a", "b", "c" microstates are shown in the context of the tICA space of the L302A system. Thus, as they stand, Figures 5 and Supp. Figure 9 are consistent with each other in the nomenclature of

the microstates and we would prefer not to alter this. We believe that sufficient explanation is provided in the text to connect the L302A microstates to those of the wild type.

25. Figure 7: Panel D: perhaps add a dashed line to indicate the suggested upward movement of TM3. Legend of panel G: To avoid confusion stick to pore radius or diameter. Why not show an actual superposition of WT and mutant of the groove like in panel C.

We now show in Fig. 7C a side by side comparison of the groove of WT and L302A nhTMEM16, and a superposition of the two was added to Supp Fig. 12I.

26. Figure 8B: Here (or at least as supplementary) would be nice to see a superposition of the cryo-EM to the nanodisc Ca-bound intermediate nhTMEM16 structure (6QMA).

Following the suggestion we have revised Supp. Figure 15, so that panels A-D now show the structural alignment of the groove region in the L302A cryo-EM structure, in the nhTMEM16 structures reported by Kaleinkova et al (6QMA, 6QMB, 6QM9), and in mTMEM16A model (5OYB).

27. Figure 9A: for a proper comparison it would be better to show the exact same (superimposed) view of the different models

This change has been made to Figure 9A.

28. S1: see major comment on figures and how to best represent the microstates. Panel C: good example on what is actually 100% in the plotted frequency? About how many structures with broken EC of how many structures in total are we talking about?

As we did for the other figures, we added in Fig. S1 the TM4/TM6 labels. Regarding the population of states and statistics of the EC gates, please see our responses to Questions 10 and 14 above. To clarify here, we make the comparison within the trajectory as described for panel C in the legend of Fig. "...compares the fractions of trajectory frames in the respective microstate in which the three functional EC gates, T333-Y439, E313-R432, and E318-R432, are simultaneously broken (the T333-Y439 gate was assumed to be broken if the T333-Y439 distance was $> 8\text{\AA}$; the E313-R432 and E318-R432 gates were assumed to be broken if the distance between carbonyl oxygen of Glu and sidechain nitrogen of Arg was $> 6\text{\AA}$)."

29. S2: Don't show the lipid in the same color as the residues. If these plots show the 7 events of flipping and it microstate 5 has been introduced as the only fully-open state that can actually flip lipids, how does it come that microstate 5 seems only to be reached in 4 out of the 7 trajectories? Indicate in figure legend as for S4 that the trajectories are plotted on the 2D landscape shown in Fig.1. S3, why not write the actual residues on top of each column? Include threshold on when something is

considering interacting/closed or broken. Why are the T33-Y439 and E313/318-432 distances not included?

We have modified the structural models in Fig. S2. As to the question about Microstate 5, we have discussed this in depth in our previous manuscript (Lee et al, Nat Comm) and only briefly summarized it in the current manuscript. To clarify here, in the ensemble simulations we have observed 7 events of lipid translocating from the IC to the EC side and in the process geometrically flipping its orientation. However, not all the trajectories were long enough to have the lipid flip to the parallel with the membrane orientation and diffuse away from the protein and into the bulk membrane (we call this “complete flip”). As detailed in Lee et al., and summarized in the current manuscript as well, this last step in the flipping process requires simultaneous breaking of the EC gates. As shown in Fig. S1C, the frequency of the EC gates simultaneously breaking is the highest in Microstate 5. Thus, this Microstate represents the states corresponding to “complete flip”. We hope this clarifies the question.

As to the T333-Y439 and E313/E318-R432 distances, we again stress that all this has been shown and described in detail in Lee et al. In the current manuscript we focused on the identification and validation of the newly described occluded state and trust that the important details described in Lee et al., can be followed together with the present results for a more complete picture.

30. S4: Why not show it for all 7 events where an occluded state was observed? For WT6 and WT4 (panels b,c,e,f) the V337-447 distances do not reach below 8Å which was the indicated hallmark for microstate 3, are they really microstate 3 occluded then? Also see comment in major remarks about observed results on proximity of lipids and reaching microstate 4. Does not look consistent and would be interesting to see how the other 3 trajectories of missing occluded states would look like.

Indeed, as described in the response to Question 1 above, we have now included all seven trajectories where an occluded state was observed. We hope that these plots together with our response to Question 1 are satisfactory for this point.

31. S5: Probably my problem but I do not understand what fraction of lipid penetration means? How deep it is in the cavity?

In Fig. S5A we plot the fraction of lipid atoms that have penetrated into the groove, i.e. what portion of lipid tail is inserted into the groove. A value of “1” means that the entire lipid tail (all its atoms) are inserted into the groove. We have modified the figure caption to clarify this point further.

32. S6: include a line indicating the >3.5Å which the authors defined as broken interaction. Panel C: Why not include here also directly the same plot for L302W and L302A? The characteristic broken TM3-4 interaction claimed to include the occluded state is stronger represented in microstate 7 and to some extent 6. Any comments on this?

A line indicating 3.5Å threshold is now included in the figure but we would like to keep the panels as they are since this figure is meant to indicate how the 302-343/347 lock is changing. As to

Microstate 7, all our results indicate that there is a strong lipid penetration into the groove in this ensemble of states. However, the mode of lipid tail insertion is somewhat different from that seen in Microstate 3. This is illustrated in Figure 2 below.

Figure 2: Comparison of modes of lipid tail insertion in Microstates 3 and 7.

As can be seen, in Microstate 7 the lipid is inserted higher up in the groove so that its backbone region is situated on the level of V337-447. This sterically restricts the middle region of the groove to remain relatively open. Figure 2 also shows that the two modes of lipid tail insertion differ in relation to the conformational state of residue Y439. Thus, in Microstate 7, the Y439 sidechain moves away from TM4 and faces towards the EC. This conformational switch is enabled by its interactions with nearby lipid head-groups (see pink headgroup and strings representing the tails of this lipid in Figure 2). We hope that this description clarifies the issue.

33. S8: Panel A: please include also a 90degree rotation as shown for the WT to see how TM4 bents. Panel E: I understand from the graph that actually 70% of L302W do not have any lipid tails inserted, which is the opposite of what is stated in the manuscript. I am reading the graph wrong?

Following the suggestion, we now include 90-degree rotation in panel D of this figure. We have also enhanced the figure with quantification of the bend angle as a function of time (panel E). As to the former panel E (now panel G), we should have labeled the figure clearer. What is plotted here is the frequency of frames with a specific fraction of lipid tail atoms inserted, i.e. number of lipid tail atoms inserted divided by the total number of lipid tail atoms (same as in answer to point 31, above). A value of "1" represents a frame in which the entire lipid tail is inserted in the groove. With this definition in mind, the figure conveys the result that the frequency of extensive lipid tail penetration into the groove is higher for the L302W mutant compared to the wild type system.

34. S11: The gel with L347 is missing, consider including for consistency

Figure S11 has been revised to include a gel with the reconstitution of L347.

35. S12: Panel H. Perhaps also show a zoom-in by 9- degree rotated superposition of the groove in both structures.

The suggested panel has been added to Figure S12I.

36. S14: meshes are barely visible.

This has been corrected in figure S14.

37. S15+16: be more precise in figure legend describing which nhTMEM16 structure was used (Ca-bound in nanodiscs, intermediate, etc...)

This is done. Please note that Supp. Figure 16, that described an 'open' model of TMEM16A, has been removed from the manuscript as suggested by the Reviewer (Question 7 above).

Once again, congratulation on the new study and I hope my comments help to further strengthen the manuscript.

Reviewer #2 (Remarks to the Author):

This paper from the Weinstein and Accardi labs seeks to understand the structural features that determine the ion and lipid conducting properties of the TMEM16 family of proteins. While there is growing interest in this family of proteins because of their disease relevance, the basic biophysics of these proteins remains enigmatic partly because of their dual function as lipid scramblases and/or ion channels. Making very impressive use of molecular dynamics simulations, biochemical analysis of lipid scrambling and ion flux, and cryo-EM these investigators provide fundamental insights into this problem. The work is truly a tour-de-force. The paper is dense with data that is interpreted fairly to yield a solid conclusion. The paper is remarkably clearly presented, especially considering the density and complexity of the data. I am not qualified to evaluate the details of the tICA approach, but the overall design seemed reasonable.

We are grateful to the Reviewer for this positive evaluation of our manuscript.

1. The authors could strive to highlight the broader significance of their work. As it stands, the details are the strength of the paper, but they might not be so interesting to those outside the small group of biophysicists who work on these TMEM16 proteins. The broader implications are relatively obvious and could be developed without much difficulty.

We thank the Reviewer for this suggestion and have added text to the Discussion and modified the Conclusions to address this helpful comment.

2. I was not convinced by the idea that lipid tail insertion was a trigger for the conformational change to the ion-conducting state. It seems that this could be presented more clearly and the delineation between correlation and causality addressed more rigorously. In Fig. 2B,C, the correspondence between lipid

insertion and conformational change is poor in the time domain. It might help to know more atomistic details about what exactly is happening during lipid insertion. If the transition to microstate 3 occurs after the lipid has departed, what evidence is there that the lipid has any role in this process? Supplementary Fig. 5 seems a crucial issue, but I felt that these data could have been crunched in a more convincing way. It would also help to see representative images of the lipid tails and water in the conduction pathway during the trajectory to microstate 3.

We thank the Reviewer for this comment that echoes Reviewer 1 as well. In our response to Question 1 of Reviewer 1, above, we have addressed this issue extensively. The full response is given above but we reproduce here some of the response to emphasize briefly that

1) ... data from the MD simulations [show] that in every instance of the observed groove closure the lipid tail had penetrated into the groove, i.e., groove closure is not observed unless a lipid tail had been inserted there first.

2) To illustrate this point better, we have revised Supp. Figure 4 so the manuscript now describes the results from all 7 trajectories pertinent to this point.... Note that in some cases the groove closure is transient (panels J-R), and in some simulations the occlusion takes place during the final stages of the trajectories (panels B, E, H, and C, F, I).... As can be seen in Suppl. Figure 4, all the trajectories progress dynamically towards Microstate 3. Furthermore, in all cases, even when the occlusion is transient, the hydrophobic tails of a single lipid molecule coordinate V337 and V447 residues. As we explain in the manuscript, physics considerations support the inference that the presence of the lipid tails can facilitate attractive interactions between the two hydrophobics, V337 and V447, thereby promoting the groove closure.

3) We also note that it is not the case that a groove closure occurs every time a lipid tail inserts. But, as stated, we have not observed an event of groove closure without the lipid tail being inserted in the groove. Nevertheless, we agree with the Reviewer that even taken together, these observations do not prove our stated conclusion definitively, as they do not completely rule out the possibility that the occlusion takes place without lipid participation and that the lipid tail insertion is not required per se but could serve as a facilitator of the closure. Therefore, we have revised the manuscript accordingly throughout to address this point (e.g., see page 23).

We hope that the revised text (in which we tone down the relationship between lipid tail insertion and the groove closure), data in revised Supp. Fig. 4, and the additional Supporting Movie provide sufficient detail to alleviate this concern by illuminating the basis of our inference and its caveats.

3. The authors sweep under the rug the fact that the ion flux assay measures Cl conductance but yet the molecular models do not have a pore dilated enough to conduct Cl. This discrepancy is brushed aside by saying that there is probably another state with a slightly dilated pore. While I agree that this is a logical presumption, it deserves more than a sentence in passing. Is there any evidence for the existence of such a state in the long MD simulations?

As we describe in the Discussion section (also see Supp. Fig. 16 of the revised manuscript), in our 15 microsecond MD simulation of the wild type nhTMEM16, which assumed the occluded state, we observed Cl ion penetration from the IC side all the way towards the EC gates. Based on the EC gating mechanism data in our previous manuscript (Lee et al.), the process of permeation did not proceed further, likely because in these simulations the EC gates remained closed. We also note that in our MD simulations of the L302A mutant the groove samples conformations that are wide enough to allow Cl⁻ permeation. It is likely that longer timescales

and/or consideration of applied voltage across the membrane will be necessary to observe the full permeation events. It will be indeed interesting to pursue this point further in future studies.

Reviewer #3 (Remarks to the Author):

The authors explore the relationship between ion conduction, lipid flipping, and protein conformational state in the fungal nhTMEM16 scramblase. Based on simulations, the authors determined that TM4 can undergo a transition from a fully open structure to a partially closed structure that stops lipid flipping. They outline the importance of lipids in this event as well as the importance of specific residues, in particular L302, which helps stabilize a TM3-TM4 interaction that becomes disrupted to close the lipid groove. Based on the importance of L302 for keeping the groove open, they made the mutation to a small hydrophobic residue (L302A) and then solved the cryo-EM structure in nano-disc with Ca²⁺ bound. They observe a closed groove that still forms a pore through the membrane large enough for small ions but not exposed to lipid, as suggested by the simulations. Overall this work is impressive with a new structure, extensive simulations, and supporting functional data. I am supportive of this work, and I think that it is a step forward to understanding how scramblases also conduct ions and the conformational changes that allow one or the other to occur. That said, you will see that I have many questions, and I think that the presentation needs to be clarified in some places.

We thank the reviewer for the overall positive evaluation of our manuscript.

Major concerns

My two biggest concerns are the following:

1. The authors have not convinced me that the lipid tail insertion is the key to groove closure. I am left with the feeling that this is an anecdotal event that may or may not be the primary pathway for groove closure (and opening??). I would like to be convinced.

Thank you for the comment. We have addressed this issue extensively in response to Question 1 of Reviewer 1, and more succinctly in response to Question 2 of Reviewer 2, above. We hope that in addition to the specific considerations, the revision of the text to tone down the relationship between lipid tail insertion and the groove closure, and the data in the revised Supp. Fig. 4 and additional Supporting Movie, provide sufficient detail to alleviate this concern.

2. *Are the TM3-TM4 motions revealed by the simulations and the L302A structure on path for wild type nhTMEM16 function, or are they specific to the L302A mutant? To this end, I would like the authors to compare TM3-4 in their new structure to the new Kaleinkova Elife structures. Are they seeing the same kind of relative motions in their structures, or is TM3-4 moving differently.*

Revised Supp. Figure 15, panels A-D show structural alignments of the groove region in the L302A cryo-EM structure, in mTMEM16A and in the nhTMEM16 structures reported by Kaleinkova et al. As can be seen from these representations (and from corresponding RMSD-s we report in Supplementary Table 1), the L302A structure is most similar to the intermediate model of nhTMEM16 by Kaleinkova et al (panel A in Supp. Figure 15). However, as also described in the manuscript, there is a significant difference in the EC gate configuration, as it is opened (broken interactions) in the L302A, but is closed in the intermediate structure; this results in movement of the EC end of TM3 in the mutant away from helices TM5/TM6.

3. Table 1. While I realize that these results are from a prior publication, it is interesting that the authors observe 7 in to out flips in the ACEMD set of simulations (called WTensemble here), and no inward flips. Do they have a feeling for why this happened? Also, would the authors comment on whether they think the ACEMD dynamics engine (barostat, thermostat, other implementations) somehow is biased towards seeing flipping, while Anton2 might suppress this? Is it all related to the propensity for the groove to fully open or the lipid systems? Have the authors used NAMD, Amber or Gromacs with the CHARMM36 force field? I realize that equilibration for Anton2 was with NAMD, but these were short simulations.

We thank the Reviewer for this comment. We note that the ACEMD simulations were all started from the same initial structure (previously equilibrated with NAMD for ~380 ns), in which the lipid groove was hydrated and a lipid head-group had penetrated to about 1/3 of the groove from the IC side. With the multi-stage protocol that involved running many independent replicates with ACEMD, we then monitored the full translocation of this very same lipid towards the EC side. Thus, these simulations were designed to follow the progress of the lipid across the groove. As we described in Lee et al. manuscript, these were unbiased MD simulations that were run in sequential stages, so that the initiation of one stage was informed by the output from the preceding one. We believe that this protocol greatly enhanced the sampling of the scrambling events.

In contrast, on Anton2 (which implements the same CHARMM36 force-field but different dynamics integrator as the reviewer points out), we are accumulating long, continuous trajectories which may exhibit single occurrences of anharmonic modes of motion (e.g. insertion of lipid tail into the groove that results in dehydration of the region) hindering the chances of sampling the whole process of lipid translocation. Nevertheless, we describe in Lee et al paper one event of lipid translocation from EC to IC side that occurred in these Anton2 simulations as (see also Table 1 in the current manuscript).

Overall, we share the sentiment of the reviewer that it is in general important for the field to have a comprehensive comparison of the results obtained with different integrators, force-fields, and equilibration strategies when simulating membrane-protein systems and to understand where the differences may lie. At this point, we think that combining very long (but few) simulations run on Anton2 with multiple relatively short simulations (obtained either with ACEMD, or another integrator) is the most powerful way to look into molecular mechanisms that are of interest. It is therefore of great interest to us, and quite reassuring, that the sequence of EC gating events, as well as the mode of opening of the mid-groove constraint, are recapitulated in the different simulations and in the expected (opposite) direction. The same pertains to the observed structural and dynamic effects of lipid tail insertion in the different simulations using the different engines and computers.

4. The authors use distances between residues in the extracellular site and residues near the mid-plane of

the groove (distance between TM4 and TM6) to judge the openness of the groove. Figure 2A shows the first two tICAs – how do these project onto your original set of CVs? How correlated are your CVs? I expect that they are highly correlated. For instance, is tICA one the extracellular site opening and tICA 2 the mid-plane opening?

We appreciate this comment. Indeed, there are some correlations among the CVs, which is as expected. In Figure 3 below we show contributions of each CV to the first tIC vectors. As can be seen, tIC2 mostly encodes for Y439 dynamics, whereas tIC1 informs on the E313/E318-432 gates. Both tIC-s have some contribution from the 337-447 distance (TM4-TM6 MID) with tIC1 showing larger contribution. While there is some correlation in CVs we selected to follow the structural changes associated with the groove dynamics and lipid translocation, our aim here was to identify major conformational states of the system rather than building kinetic models based on the population of states (for which orthogonality of CVs would have been one of the key requirements).

Figure 3: Contributions of the CVs used for tICA analysis to the first tIC vectors.

5. I very much like the kind of analysis that the authors present in Figure 2; however, I have a hard time teasing apart the lipid degrees of freedom from the protein degrees of freedom. For instance, the authors discuss a lipid flipping event and the corresponding microstates in Fig 2A, but I got lost very quickly about whether this was a discussion about 1 lipid flipping event or a description of many events that all behave similarly. How biased are the protein CVs by where permeating lipids are? To this end, are the CVs computed with c-alpha-c-alpha distances or side chain distances (I am pretty sure they are all c-alpha-c-alpha)? Nonetheless, side chain distances would constitute a large motion in tICA space during a lipid permeation event that separates residues (say at the extracellular site), while having very little influence on the slower degrees of freedom of the protein (i.e. helix-helix separation).

The basic analysis of the lipid flipping process and its relationship to the EC gate dynamics was presented in great detail in our recent paper (Lee et al., Nat Comm), given the similarity of the system and the process we only briefly summarized in the current manuscript how the new tICA space presented in this work recapitulates all the major steps we described in Lee et al. paper. To clarify for the Reviewer, the lipid flipping pathway on tICA space described in the current manuscript describes many (7 to be precise) events of lipid flipping all behaving similarly.

Because the dynamics of the protein and the permeating lipids are mechanistically related to each other, the CVs we chose to describe our system will implicitly include information regarding the positioning of the permeating lipid. We agree with the Reviewer that sidechain dynamics are essential here. Consequently, as we describe in Methods, the CVs designed to follow the dynamic rearrangements of the groove during lipid translocation were defined in the following manner: (1)-the minimal distance between T333 and Y439 (i.e. the distance between the closest pair of atoms on the two residues); (2)-the minimal distance between Y439 and R432; (3-4) distance between E313 and R432 and between E318 and R432 (defined as distance between carbonyl oxygen of Glu and sidechain nitrogen of Arg); and (5)-the Ca-Ca distance between V337 and V447 residues. Thus, for all the CVs except the V337-V447 distance, we chose the description that is based on the sidechain dynamics. For V337-V447 the magnitude of change from the open to occluded state was relatively large and the key contribution had to do with spatial reorganization, so that Ca-Ca distance was sufficient to describe this dynamic change.

6. The authors explain at the top of page 9 how extracellular residues E313 and R432 play an important role in lipid flipping and release, and they cite a past paper where they discuss these residues and another one. They should also cite the Bethel and Grabe PNAS paper, where these two first residues were first highlighted.

Thank you for the suggestion. This lapse was corrected, and the citation included in the revised manuscript.

7. From a structural point of view, the movements of TM3 and TM4 in Figure 1A in microstates 1 versus 3 are very appealing. This is how I imagine lipid gating likely happens. In the Markov State Modeling literature, is it better to call these states macrostates rather than microstates? I do not remember the criteria used by the Pande lab (and others) to distinguish these two ideas, but it is worth looking into.

Thank you for this comment. It is somewhat arbitrary, from our reading of the Markov State Modeling literature, when are the states called microstates or macrostates. Typically, the population of macrostates is at least one order of magnitude or so larger compared to microstate. For example, in our earlier studies of transporters and GPCRs, we would discretized the tICA space in 100 microstates which we would then reduce to 12-15 macrostates. In the current study, we discretized the tICA space in 50 clusters, which is somewhere in between. Usually, the further clustering into macrostates is useful for visualization of paths from transition path theory analysis, whereas the microstate formalism is preferred for quantification of kinetics.

8. Several places through the manuscript the authors use the word enhanced. I do not see how these are enhanced simulations in the classic/standard view of enhanced sampling. I would remove this term, or if I missed it, please correct me.

We appreciate this observation and have revisited the text and revised it accordingly. The only instances where we maintain the word “enhanced” are in the phrase “*enhanced complement of trajectories*”, which refers to the large number of trajectories (as opposed to the initial, Anton 2

generated small number of long ones) without implying that these trajectories were accumulated using enhanced sampling techniques.

9. In Figure 2A, please provide the axes values so that the reader can appreciate where these points lie in the phase plane of Figure 1A. This figure is very interesting, and I find this kind of molecular level detail explaining a configurational transition enlightening; however, this is all based on a single trajectory. There is nothing to suggest that this is the primary pathway to groove closure. It is said that microstates 4 and 7 experience tail entry in several cases, but how many other examples like WT6 are there? Can you provide statistics about pore closure? If not, the authors should present some kind of a PMF or energy calculation to support the generality of their claim in Figure 2.

As requested, we have added axes to Figure 2A.

Regarding the rest of the comment, we have clarified above that we now show all 7 instances of the groove closure (6 of them shown in Supp. Figure 4, and one in Fig 2).

In relation to the groove closure mechanism, we have addressed this issue extensively in response to Question 1 of Reviewer 1 and Question 2 of Reviewer 2. We explain our reasoning about the involvement of lipid tail insertion in this mechanism. We also stress that we cannot rule out another mechanism of the groove closure that does not involve lipid tail insertion for which, however, there is at the moment no indication. We hope that the revised text (in which we tone down the relationship between lipid tail insertion and the groove closure), together with the revised Supp. Fig. 4 and additional Supporting Movie provide sufficient detail to alleviate this concern.

10. Figure 3 is not really described in the main text. Is panel C defined by the box in panel A? Likewise, are panels B and D above and below this boxed region?

We assume the Reviewer meant Figure 4 (water count plots) not Figure 3. The reviewer is correct – panel C describes the water count in the central region of the groove whereas B and D show the same measures above and below this boxed region. As we realized that for the purpose of the narrative in the manuscript, only the distributions in the central region of the groove is relevant, we decided to eliminate panels B and D from this figure.

11. The simulations in Figure 4 suggest that partial closure of the gate is accompanied by separation of L302 on TM3 from 343/347 in TM4. The authors should look at the homologous residues across all known structures (scramblers and not as well as their new structure reported here) and show that partial of full closure of the groove does involve greater distances between these residues. This is important because it gets at whether TM3-4 do or do not move as a unit during gating. I now realize that this is done later in manuscript with regard to the new mutant structure – very nice, but I would like to see it compared to all structures.

We appreciate the comment. We have now added structural alignment data for the groove region of the L302A cryo-EM model with all the available structures of Ca-bound nhTMEM16, afTMEM16, mTMEM16A, mTMEM16F and hTMEM16K. To this end, we used the Pymol “align” utility. From this alignment we identified residues that structurally align with L302 and I343 positions in these structures and calculated Ca-Ca distances. The comparison of these distances

is presented in the Table below (shown in the table is the listing of residues chosen for this analysis in the respective structures). As can be seen, the distance is relatively large for the closed and intermediate structures and shorter for the open structures. The only outlier in this trend appears to be hTMEM16K in which the Ca-Ca distance does not seem to change much; it is noteworthy, however, that the rearrangements seen in this homologue differ slightly from those seen in aTMEM16 and nhTMEM16 as the TM3-TM5 region rotates together by $\sim 17^\circ$ relative to the scaffold region (Bushell et al., BioXRiv, 2018), whereas in the fungal homologues TM4 moves away from TM3 and TM5 (Falzone et al., Elife, 2019; Kalienkova et al., Elife, 2019).

mTMEM16A (closed, 5OYB)	L509, L553	10.6 Å
nhTMEM16 (closed, 6QMB)	L302, I343	9.5 Å
mTMEM16F (closed, 6QPC)	I474, L524	9.3 Å
hTMEM16K (closed, 6R7X)	S329, M373	8.3 Å
nhTMEM16 (intermediate, 6QMA)	L302, I343	9.2 Å
nhTMEM16 L302A (intermediate, 6OY3)	A302, I343	8.9 Å
hTMEM16K (open, 5OC9)	S329, M373	8.5 Å
aTMEM16 (open, 6E0H)	L294, M335	7.8 Å
nhTMEM16 (open, 6QM9)	L302, I343	7.4 Å

While overall, this analysis indeed suggests that TM3-TM4 distance grows as the groove transitions from the open to intermediate or closed states, we believe that comparison of these static structures is not very informative since we find, as described in the manuscript, that TM3-TM4 segments of the protein are very flexible. Thus, robust structural comparison will entail accumulation and detailed analysis of extensive molecular dynamics trajectories of these models which is the topic of a separate study.

12. The experiments in Figure 5 are impressive. Were any mutants at V447 made (here or in other studies), and if so, what do they do? Is the scramblase activity as critical at this site?

We assume that Figure 6 is meant here, and we thank the Reviewer for this positive comment. We are not aware of any studies where V447 was mutated but we have reported results on the V337W mutant (Lee et al., Nat Comm) which surprisingly showed enhanced ion transport activity over the wild type under conditions of no Ca^{2+} . Other studies as well (Jiang et al., Elife, 2017; Le et al., Nat Comm 2019) show functional significance of residues on TM4 near V337.

13. On page 16, the authors state:

“The nhTMEM16 L302A/nanodisc complex is bent along the dimer cavity (Supplementary Figure 12I), consistent with recent findings on how the TMEM16 scramblases affect the surrounding membrane 19,20.”

I appreciate that the authors are referring to recent structural work in nano-disc that has experimentally revealed the shape of the surrounding membrane, but it might be appropriate to cite Bethel and Grabe as they were the first to predict this several years prior with computation.

This is absolutely correct and is included in the revised manuscript.

14. *The structural work in Figure 7 is very nice, and the figure illustrates all of the concepts discussed throughout the manuscript nicely. I appreciate the overlay of the recent structures from the Kaleinkova nhTMEM16 manuscript. One thing I noticed is that while TM6 is in a similar place in all 4 structures, the position of TM3 is similar for all except the most recent mutant. Does this lead us to think that the mutant is adopting a conformation that is not frequently sampled in the WT protein? The pore electrostatics profile work is nice. I especially like the many curves superposed on each other taken from the simulations (Fig. 9F).*

We thank the Reviewer for this comment and for expressing appreciation for our work. As described in the manuscript and hopefully now illustrated better in revised Supp. Figure 15, panels A-D, the structure of the L302A mutant indeed has a distinct structural feature on the EC side which differentiates it from the other available structures. It is important to note, that in several cryoEM structures of TMEM16 proteins TM3 is often poorly resolved, which is suggestive of it being quite dynamic; indeed TM3 was observed in multiple conformations in both TMEM16A and nhTMEM16 (Paulino et al., Nature, 2017; Kalienkova et al., Elife, 2019). Thus, we find that the EC gates, involving E313, E318 and R432 are broken in the L302A structure. As a result, TM3 is clearly shifted away from TM6 compared to its position in other models (i.e. intermediate, open, close nhTMEM16). As we describe in the manuscript (see also new Figure 10 in the revised main text), in our simulations of the wild type we observe E318 moving away from R432, but the E313-R432 interaction is still intact when the occluded conformation is formed. Our computational analysis did predict that the occluded conformation would be more frequent in the L302A mutant compared to the wild type based on our observation that the occlusion was observed in 75% of the trajectories of the mutant (3/4 protomer) whereas in the wild type it was much rarer (Table 1). This is consistent with our analysis of the L302A mutant cryoEM data where the only class going to high-resolution is with an occluded groove.

15. *The work reported here may be used to support a model that ions and lipids permeate via different states. The authors don't make any comment about lipid-ion co-permeation as shown in the Elife paper by Tajkhorshid and Hartzell. I think that the authors should at least comment on this even if they can't make any solid claims one way or the other. In particular the 3 models put forth at the end of the Kaleinkova Elife paper are useful for framing this.*

Our findings do not rule out that in nhTMEM16, ion translocation events can also occur through the scrambling-competent conformation, as seen in the Elife paper by Tajkhorshid and Hartzell. What our results suggest is that the occluded state we describe in our paper can still conduct ions but not lipids. We have revised the manuscript (page 23) to include discussion of the Elife paper that the reviewer mentions.

Minor

16. *At the top of page 4, first line. "it occupies a more central" – what is "it"? Are you referring to Ca²⁺ here?*

By "it" we meant TM6 and this is clarified in the revision. Thank you.

17. *First paragraph of figure 4:*

"...giving rise to the Ca²⁺-bound closed conformation seen in the TMEM16K and TMEM16F scramblases 21". Reference 21 is for K only, not F. Please insert F citation.

Done with thanks.

18. *On page 4, the authors state:*

*"In the TMEM16A chloride channel, the groove serves as an ion-selective pore ..."
I would not call this a groove, and I think that many others would not either. I think some care should be taken at a few points in the manuscript to not call pores grooves.*

We modified this sentence as follows: "In the TMEM16A chloride channel, the region that is analogous to the groove in the scramblases..."

19. *Again in the middle of page 4,:*

"...wider groove compared to that in the Ca²⁺-bound TMEM16A channel"

I think that care should be given to calling structures grooves versus pores. Imagine that a conformational change opens a pore to form a groove. I still think that one should not call the pore a groove. I also realize that the authors are careful about this at other points in the manuscript.

We modified this sentence as follows: "... is characterized by a somewhat wider groove compared to the dimensions of the pore in the Ca²⁺-bound TMEM16A channel".

20. *At the bottom of Page 16 the authors use the phrase "extracellular vestibule" without defining it.*

Thank you, this is now clarified (now on page 17 of the revised manuscript).

21. Throughout the manuscript "microstate" and "alanine" are often capitalized when they should not be. For instance on page 24, "L302 to Alanine" should be "L302 to alanine".

Corrected; thank you.

Reviewers' Comments:

Reviewer #1:

Remarks to the Author:

I believe the authors have addressed all comments and provided a thorough and improved revision. I strongly favour publication at Nature Communication.

Best,

Reviewer #2:

Remarks to the Author:

The authors have answered all of my scientific questions, but I think the paper needs a little work to make it more accessible to a general audience.

I sympathize with the authors' challenge of explaining these very intricate structural studies. However, I feel that the jargon/terminology is often imprecise and makes this paper difficult to follow. The terms need to be more precisely defined and/or a cartoon model figure should be shown that prepares the reader for what is coming. In the paragraph starting with line 74, the authors use the terms "open" and "closed" to define the relationship of TM4/6 to the bilayer. But, later, the terms "closed" and "occluded" are used interchangeably (for example, page 11-12, lines 246-250 "occlusion of the groove occurs and groove closure occurs" and Figure 5 legend and text). Because the terms "open" and "closed" could be misconstrued to be synonymous with conductive/non-conductive, and because both ion and lipid conduction (which are evaluated by different techniques) are involved, the use of the binary terms "open" and "closed" is confusing. I would advocate eliminating the use of "open" and "closed" entirely or at least to define precisely – in Angstroms or picosiemens – what these terms mean. The distinction becomes crucial in line 123 when the groove is "closed" (= non-conducting) to lipids but "wide" enough (does that mean "open" to the membrane?) to conduct ions and in line 128 referring to an "open ion-conducting" (open to the membrane or open = conducting?) conformation. Another confusion is introduced by the terms "occluded", "occluded groove", and "intermediate" because it is not immediately clear until later whether these are the same and how they relate to open/closed/conducting/non-conducting.

Line 43. Improperly placed comma.

Line 60. "On the opposite sides of the dimer interface." The wording is imprecise.

Line 141-2. "insertion of a lipid tail into the groove which promoted the sampling of conformations" The grammar gives this sentence an odd meaning.

Table 2. Seems to be missing some numbers or the legend "listing of the pair-wise distances" is wrong.

Line 188. Capitalize first letter of sentence.

Line 506. conductive

Reviewer #3:

Remarks to the Author:

I think that you have done a terrific job of addressing all of the questions posed by the reviewers - very very nice paper.

Reviewer 1 (Remarks to the Author):

I believe the authors have addressed all comments and provided a thorough and improved revision. I strongly favour publication at Nature Communication.

We thank Dr. Paulino for this evaluation.

Reviewer 2 (Remarks to the Author):

The authors have answered all of my scientific questions, but I think the paper needs a little work to make it more accessible to a general audience.

We thank the reviewer for the overall positive feedback.

I sympathize with the authors' challenge of explaining these very intricate structural studies. However, I feel that the jargon/terminology is often imprecise and makes this paper difficult to follow. The terms need to be more precisely defined and/or a cartoon model figure should be shown that prepares the reader for what is coming. In the paragraph starting with line 74, the authors use the terms "open" and "closed" to define the relationship of TM4/6 to the bilayer. But, later, the terms "closed" and "occluded" are used interchangeably (for example, page 11-12, lines 246-250 "occlusion of the groove occurs and groove closure occurs" and Figure 5 legend and text). Because the terms "open" and "closed" could be misconstrued to be synonymous with conductive/non-conductive, and because both ion and lipid conduction (which are evaluated by different techniques) are involved, the use of the binary terms "open" and "closed" is confusing. I would advocate eliminating the use of "open" and "closed" entirely or at least to define precisely – in Angstroms or picosiemens – what these terms mean. The distinction becomes crucial in line 123 when the groove is "closed" (= non-conducting) to lipids but "wide" enough (does that mean "open" to the membrane?) to conduct ions and in line 128 referring to an "open ion-conducting" (open to the membrane or open = conducting?) conformation. Another confusion is introduced by the terms "occluded", "occluded groove", and "intermediate" because it is not immediately clear until later whether these are the same and how they relate to open/closed/conducting/non-conducting.

We have revised the text so that the structural states are now clearly defined in the Introduction and consistently used throughout the manuscript. These states are designated as follows: Ca²⁺-free closed, Ca²⁺-bound closed, Intermediate, Ion-conductive, Membrane-exposed, and Lipid-conductive.

Line 43. Improperly placed comma.

Line 60. "On the opposite sides of the dimer interface." The wording is imprecise.

Line 141-2. "insertion of a lipid tail into the groove which promoted the sampling of conformations" The grammar gives this sentence an odd meaning.

Table 2. Seems to be missing some numbers or the legend "listing of the pair-wise distances" is wrong.

Line 188. Capitalize first letter of sentence.

Line 506. conductive

Thank you for pointing these out. We have addressed all these issues in the revision. Note, that due to space limitations some of the above Lines may have been modified, removed from the manuscript or moved to the Supplementary Information (such as Table 2 which is now Supplementary Table 2).

Reviewer #3 (Remarks to the Author):

I think that you have done a terrific job of addressing all of the questions posed by the reviewers - very very nice paper.

We thank the reviewer for this evaluation.